# Disrupted host-microbiota crosstalk promotes nonalcoholic fatty liver disease progression by impaired mitophagy

Wenjing Yin,[1] Wenxing Gao,[1] Yuwei Yang,[1] Weili Lin,[1] Wanning Chen,[1] Xinyue Zhu,[1] Ruixin Zhu,[1] Lixin Zhu,[2] Na Jiao[3]

**ABSTRACT** The intricate interplay between host genes and intrahepatic microbes is vital in shaping the hepatic microenvironment. This study aims to elucidate how host-microbiota interactions contribute to the progression of nonalcoholic fatty liver disease (NAFLD). Hepatic gene and microbial profiles were analyzed from 570 samples across five cohorts, including 72 control, 124 nonalcoholic fatty liver (NAFL), 143 Borderline, and 231 nonalcoholic steatohepatitis (NASH) samples. Least absolute shrinkage and selection operator penalized regression and sparse canonical correlation analysis were utilized to identify host-microbiota interactions and their function. Validation was performed using a bulk transcriptomic data set comprising 1,332 samples and a single-cell transcriptomic data set of seven samples. We observed stage-specific gene expression changes of disrupting energy metabolism and immune responses, alongside microbial shifts shaping the NAFLD microenvironment. Additionally, we identified 5,537, 1,937, 1,485, and 2,933 host-microbiota interactions in control, NAFL, Borderline, and NASH samples, respectively. *Escherichia coli* and *Actinomyces naeslundii* dominated the interaction network in control but were replaced by Sphingomonadales and Sphingomonadaceae in disease stages from NAFL, preceding the transcriptomic tipping point observed in Borderline. In NASH, interactions significantly weakened, accompanied by the loss of mutualistic interactions between bacteria such as Bacillales, *Ralstonia insidiosa*, Sphingomonadaceae, and host mitophagy genes including *SQSTM1*, *OPTN*, and *BNIP3L*. Single-cell data sets confirmed these interactions were co-localized in macrophages and monocytes in control, which shifted to hepatocytes and endothelial cells in NAFLD. Shifts in host-microbial interaction signal early microenvironment changes. Disturbed host-microbiota interactions impacting mitophagy can trigger a pro-inflammatory hepatic microenvironment, potentially driving disease progression.

**IMPORTANCE** This study integrated multiple cohorts to uncover fundamental and generalizable signals in the progression of nonalcoholic fatty liver disease. Key changes in both liver gene expression and microbiota were identified across disease stages, with microbial composition and interactions with host offering earlier insights into microenvironmental changes. Notably, host-microbiota interactions related to mitophagy, crucial in early stages, were destroyed in nonalcoholic steatohepatitis. This disruption may contribute to the worsening inflammation and disease progression.

**KEYWORDS** intrahepatic microbiome, hepatic microenvironment, host-microbe interaction, nonalcoholic fatty liver disease, mitophagy, disease progression

Nonalcoholic fatty liver disease (NAFLD) is a chronic liver disorder encompassing a spectrum of stages (1). The initial stage, nonalcoholic fatty liver (NAFL), is characterized by excessive triglyceride accumulation in hepatocytes (2). In contrast, the advanced stage, nonalcoholic steatohepatitis (NASH), involves irreversible liver injury and severe hepatic inflammation (2). Patients with histological features intermediate

Address correspondence to Ruixin Zhu, rxzhu@tongji.edu.cn, Lixin Zhu, lixinzhu@imu.edu.cn, or Na Jiao, najiao@fudan.edu.cn.

Wenjing Yin and Wenxing Gao contributed equally to this article. The author order was determined by drawing straws.

The authors declare no conflict of interest.

See the funding table on p. 17.

between NAFL and NASH are classified as Borderline steatohepatitis (Borderline) (3). The transitions between these stages are complex, with the mechanisms driving progression and potential regression remaining unclear (4).

The gastrointestinal tract harbors the most abundant and diverse microbial community in the human body. Under physiological conditions, these microbes are typically confined within the gut lumen by an intact intestinal epithelial barrier (5–7). However, certain microbes and microbial products can enter systemic circulation, reaching distant tissues such as the liver, primarily through portal circulation (8). This process of microbial translocation is notably enhanced under pathological states, such as chronic liver diseases including NAFLD, due to increased gut permeability and impaired barrier integrity (7, 9). Consequently, the liver becomes a secondary reservoir for a unique microbial community, as evidenced by sequencing (10–12), quantitative polymerase chain reaction (qPCR) (13), and fluorescence studies (14). These hepatic bacteria are closely associated with the liver microenvironment such as fibrosis (11) and inflammation (14). These findings emphasize the importance of directly investigating the hepatic microbiota, which may differ substantially from the gut microbiota due to selective translocation and local hepatic immune interactions. However, such specific roles and mechanisms of hepatic microbes remain largely unexplored.

Multiple factors act in concert in NAFLD pathogenesis (15, 16), including genetic abnormalities (16), metabolic dysregulation (17), and gut microbiota dysbiosis (18–20). Therefore, to understand disease mechanisms comprehensively, it is essential to investigate the interplay between these risk factors (21, 22). For example, evidence suggests that human genetic variation shapes the composition of hepatic microbiota. These microbes further influence NAFLD progression through bacterial metabolites (15). While NAFLD progression typically follows a continuum from control to NAFL, Borderline, and NASH (1), how these host-microbiota interactions evolve and influence this trajectory remains unknown.

Building upon emerging evidence of microbial involvement in liver pathophysiology, several critical knowledge gaps remain in our understanding of NAFLD. First, the composition and dynamics of hepatic microbiota across distinct disease stages have not been systematically characterized. Second, patterns of host-microbe interactions and their functional implications in liver disease remain poorly defined. Third, the mechanistic roles by which hepatic microbiota influence host metabolic and inflammatory pathways during disease progression are still unclear. Our study aims to address these gaps by collecting transcriptomic data across the full spectrum of NAFLD progression. Five cohorts of 570 samples from diverse populations were combined into a discovery cohort. A mega-analysis-like integrated approach was used to identify universally pathological signals (23–25). Our findings revealed increasing disruption of hepatic microbiota composition and host-microbiota interactions during NAFLD progression. This disruption particularly involved pathways related to mitophagy, which may further lead to energy metabolism and abnormal immune dysregulation. We validated these results in an external bulk and a single-cell cohort. Collectively, this study comprehensively explores host genetics, microbes, and their interactions, providing a fresh perspective on the hepatic microenvironment during NAFLD progression.

## MATERIALS AND METHODS

### Sample collection

#### Discovery cohort collection and curation

RNA-seq data sets from liver biopsies of healthy individuals and NAFLD patients were searched and curated from public databases, including GEO, BioProject, and ArrayExpress, using keywords "RNA-seq," "transcriptomic," and "liver biopsies." To ensure rigor and consistency, only data sets meeting uniform criteria were retained, specifically those providing transcriptomic data along with clearly defined clinical group information.

Studies were excluded if they lacked clear pathological diagnoses from liver biopsies or failed quality control standards (Fig. S1). Individuals with histories of alcohol intake or viral infection were also excluded to ensure the specificity of our patient cohort to NAFLD. The final discovery cohort comprised five studies: cohort ER from Govaere et al. (26) (European Region countries, *n* = 216, PRJNA558102), cohort US1 from Yao K et al. (27) (USA, *n* = 57, PRJNA767535), cohort US2 from Pantano et al. (28) (USA, *n* = 143, PRJNA682622), cohort JP from Kozumi et al. (29) (Japan, *n* = 97, PRJNA704861), and cohort DM from Gerhard et al. (30) (Denmark, *n* = 57, PRJNA523510). Based on the NAFLD Activity Scores (NAS) system (2), samples from cohorts ER, US1, and US2 were classified into NAFL (NAS < 3), Borderline ($3 \leq$ NAS < 5), and NASH (NAS $\geq$ 5). Cohort JP was staged using the Matteoni classification (31) and cohort DM using SAF systems (32), both aligning closely with NAS criteria. Notably, among the four data sets containing control samples, cohort US1 defined control samples as healthy individuals (27), while cohorts ER, US2, and DM designated individuals undergoing bariatric surgery as control samples. It is important to note that these individuals were confirmed to have no biochemical or histological evidence of NAFLD (26, 28–30). Samples were all obtained through standardized percutaneous needle biopsies, primarily from the right lobe of the liver.

### External validation cohorts collection

A recently published large bulk RNA-seq data set comprising 1,332 samples of NAFLD patients (33) was incorporated. This data set includes Scotland samples with fibrosis stages 0–4: 68 without fibrosis and 602, 202, 228, and 232 in fibrosis stage 1, 2, 3, and 4, respectively. In addition, a single-cell RNA-seq data set consisting of five healthy control samples and two NAFLD patients (34) was included for validation at a higher resolution.

## Hepatic gene expression preprocessing

### Bulk RNA-seq data preprocessing

Low-quality sequences and unnecessary adapters were removed using Trimmomatic (version 0.39), retaining reads with a base quality score greater than 30. Next, these high-quality reads were aligned to the CHM13 human reference genome with STAR (version 2.7.10a) (35). After alignment, to measure gene expression levels, we quantified reads mapped to each gene with STAR. Gene annotations were retrieved using the R package "biomaRt" (version 2.52.0), and only protein-coding genes were retained. In addition, genes with expression variance below the 25th percentile across all the samples were excluded, as they were considered less informative. The same pipeline was performed to the bulk RNA-seq external cohort for validation.

### Batch effects assessment and management

To evaluate gene expression profiles across batches, we first assessed and visualized potential batch effects using principal coordinates analysis (PCoA), then tested their statistical significance and applied batch correction to eliminate any potential biases. Specifically, we performed PCoA using the Euclidean distance matrix of log-transformed gene expression matrix using R package "mixOmics" (version 6.22.0). The results were visualized using the R package "ggplot2" (version 3.5.1). To quantify the extent of batch effects statistically, permutational multivariate analysis of variance was performed using R package "vegan" (version 2.6-4), with "Study" as a surrogate variable for batch effects. Finally, to remove these batch effects, the "RemoveBatchEffect" function in R package "limma" was applied (version 3.54.2).

### Gene expression analysis

To accurately identify genes associated with NAFLD conditions, we first stabilized the gene expression profiling using variance stabilizing transformation, then detected

differentially expressed genes (DEGs) with R package "limma" (version 3.54.2) (34). Genes with the false discovery rate (FDR) < 0.05 and absolute log2 fold change (log2FC) >1.2 were defined as DEGs. Finally, to interpret the biological meaning behind these gene changes, we examined whether certain biological pathways were notably active or suppressed using gene set enrichment analysis (GSEA) with R package "clusterProfiler" (version 4.4.4). Pathways with FDR < 0.05 were defined as significantly enriched.

## Hepatic microbial composition preprocessing

### Extraction and annotation of candidate microbial reads

Since transcripts from microbes can also be captured by sequencing, we used the state-of-the-art software and strategies to extract the hepatic microbial sequences within liver biopsy RNA-seq data (36, 37) (Fig. S2 and S3). First, reads unmapped to the human genome by Kneaddata (version 0.6.1) were defined as candidate microbial reads. Then these reads were annotated using KrakenUniq (version 1.0.4), a tool specifically suited for low-biomass samples as it effectively differentiates genuine microbial signals from false positives caused by highly repetitive or contaminated sequences (38), via mapping to a customized database comprising the human genome, complete microbial genomes (retrieved from RefSeq 5th on September 2023, included 35,022 bacterial genomes, 14,993 viral genomes, 35 fungal genomes, and 540 archaeal genomes), and artificial sequences (UniVec and EmVec).

### Contamination removal to obtain true microbial profile

A stringent five-step protocol was applied to mitigate contaminations (Fig. S2). (i) Taxa with low unique k-mer counts (threshold: read length − k-mer length + 1) were filtered, as true microbial taxa generate evenly distributed k-mers across their genomes. (ii) Taxa with low total k-mer counts were considered random occurrences and removed (threshold: exceeding read counts by fivefold). (iii) Taxa showing any non-significant Spearman correlations among total read counts, unique k-mer counts, and total k-mer counts were removed as contaminants (39). (iv) Taxa resembling known experimental contaminant distribution were removed (39). (v) Taxa matching a published reagent contamination list were removed (40).

### Microbial profile preprocess and batch effects management

To focus on reliable microbial signatures, rare microbial taxa with a relative abundance below 0.01% in fewer than 10% of samples within each data set or those only present in a single data set were excluded. Considering the compositional characteristics of microbiome data, we then normalized the microbial profile using centered log ratio (CLR) transformation with R package "compositions" (version 2.0). Potential batch effects across data sets were systematically assessed and corrected using the R package "limma," consistent with the management for host gene expression data.

### Microbial profile analysis

To measure microbial diversity within individual samples, we calculated the Shannon diversity index on CLR-transformed abundance profiles with R package "vegan." Differences in microbial composition between samples were assessed using Aitchison distance using "vegan." To identify taxa with significantly different abundance across conditions, we applied differential abundance analysis using the Wilcoxon-Mann-Whitney test. Taxa with FDR < 0.05 and absolute log2FC > 1 were defined as differentially abundant taxa.

## Interaction analysis between hepatic genes and microbes

### Host-microbiota interactions identification

Our goal was to pinpoint microbial taxa whose abundances best predict the expression of host genes across different stages of NAFLD. To achieve this, we used least absolute shrinkage and selection operator penalized regression (Lasso) with R package "glmnet" (version 4.1-7) based on batch effect-corrected gene and microbial profiles. To capture interactions comprehensively, microbial abundance data from all taxonomic levels from phylum to species were merged into a single combined taxa matrix. The optimal tuning parameter $\lambda$ was determined through leave-one-out cross-validation. Significant interactions (FDR < 0.01) were identified using the "hdi" package (v.0.1-9). To ensure stability, the model was refitted 100 times on random subsets with perturbed $\lambda$, retaining interactions present in $\geq 60\%$ of iterations. Interaction strength was quantified through Spearman correlation via "stats" (version 4.2.1). Furthermore, to rule out potential confounding effects of factors like age, gender, and sequencing platform, multivariate linear regression models were constructed using the R package "stats."

### Interaction function annotation

To explore the biological relevance of identified host-microbial interactions, we performed sparse canonical correlation analysis (SparseCCA). This method projects both host genes and hepatic microbial data into a shared latent space, identifying correlated functional clusters of genes and microbes. SparseCCA was implemented using the R package "PMA" (version 1.1), incorporating an L1 penalty to select the most strongly associated host genes and microbes. Subsequently, to elucidate the biological functions represented by each interaction cluster, we performed the KEGG pathway enrichment of the genes involved in each cluster using the "enrichKEGG" function in R package "clusterProfiler." Pathways were considered significant if they had a corrected FDR of less than 0.05.

## Validation on single-cell resolution

### Single-cell data preprocessing

The single-cell validation cohorts included liver tissue and peripheral blood mononuclear cell (PBMC) samples from healthy and NAFLD individuals (34). First, raw sequencing data were processed using Cell Ranger (version 8.0.1) for alignment to the human genome, barcode correction, and unique molecular identifier (UMI) counting. Doublets were identified and removed using the R package "DoubletFinder" (version 2.0.4). To further ensure the integrity of liver-specific analyses, samples were integrated, and batch effects were mitigated using the R package "harmony" (version 1.0). Next, we excluded any liver-derived cells clustering closely with PBMCs, removing contamination from circulating cells and ensuring the identification of genuine liver-resident cells. Cells were clustered using the R package "Seurat" (version 4.1.0) and annotated using the R package "singleR" (version 1.8.1) using default "HumanPrimaryCellAtlasData" as the reference data set.

### Microbe-carrying cells identification

To accurately detect and annotate microbes within single-cell RNA sequencing data, we employed an optimized version of the SHAMI framework (39). Briefly, reads that did not align to the human genome were further analyzed to identify candidate microbes using KrakenUniq, an improved microbial classifier optimized for low-abundance, low-biomass microbes (38). To minimize contamination risks, candidate microbial reads were rigorously filtered using statistical correlations between total k-mer counts, unique k-mer counts, and total read counts. Additionally, we removed potential laboratory contaminants by excluding reads matching a laboratory-specific blacklist (40). These filtered microbial reads were linked back to individual cells based on cell barcode information,

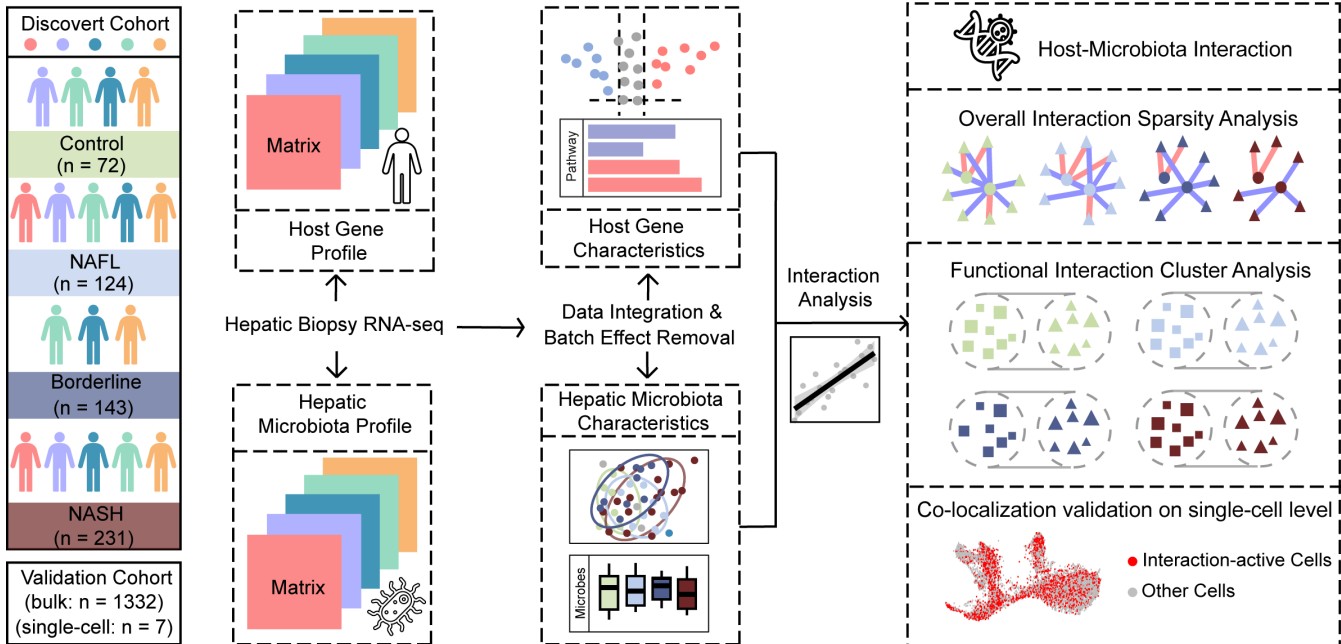

**FIG 1** Study design. This study encompassed samples covering the full spectrum of NAFLD, including control, NAFL, borderline, and NASH. Host gene profiles and paired hepatic microbial profiles of each data set were extracted and annotated from RNA sequencing data of liver biopsies. Integrated analyses were conducted to establish a unified representation of features at both host gene and hepatic microbiota level. Subsequently, host-microbiota interactions were delineated from both overall sparsity level and functional cluster level across stages, aiming to understand their contribution to the hepatic microenvironment. Validation was performed on external bulk and single-cell RNA data sets, while single-cell data facilitate the co-localization of host-microbial interactions at a finer resolution.

classifying cells containing at least one microbial read as "microbe-carrying," while those with no microbial reads were classified as "microbe-free."

### Host-microbiota interaction validation

We next validated predicted interactions between host genes and microbes by assessing their spatial co-localization within individual cells. For each interaction cluster identified by SparseCCA, gene expression and microbial abundance were evaluated at the single-cell level. Cells were defined as active interaction cells if they contained at least one gene and one microbe whose expression levels within the cluster exceeded the 75th percentile threshold. We visualized these active cells by highlighting them in red on a uniform manifold approximation and projection plot, with inactive cells displayed in gray. This visualization allowed us to clearly observe the spatial distribution and potential interaction patterns at single-cell resolution. All single-cell analyses and visualizations were conducted using the R package Seurat.

## RESULTS

### Study design

To investigate core hepatic microenvironment mediated by host-microbiota interactions across the NAFLD spectrum, we conducted an integrated study leveraging large-scale data mining (Fig. 1). The discovery cohort included 570 samples from five data sets spanning regions such as the US, Japan, and Europe (45.45% male, age: 41.00 ± 18.92 years for males, 44.83 ± 17.88 years for females, based on available data, Table S1). Samples were classified into 72 control, 124 NAFL, 143 Borderline, and 231 NASH using criteria detailed in Materials and Methods (Fig. S1). This pooling strategy ensured identification of fundamental and universal pathogenic signals while minimizing cohort-specific artifacts.

After integration and batch effects correction (Fig. S2 to S5), expression profiles of 14,414 protein-encoding genes and abundances of 350 microbial taxa were quantified. Genes and taxa associated with disease progression were identified, and their interplay was analyzed to uncover potential mechanisms driving disease progression.

To validate key findings, we utilized a recently published bulk RNA-seq data set comprising 1,332 samples as an external validation cohort to confirm the hepatic microbial composition and host-microbial interactions. Additionally, a single-cell RNA-seq data set of seven liver samples ($n = 5$ healthy, $n = 2$ NAFLD) was reprocessed to investigate microbial-carrying cells and the host-microbial interactions at finer resolution.

## Altered gene expression suggested disturbed energy metabolism and active immune response in NAFLD

To uncover host transcriptional changes associated with NAFLD progression, we first confirmed high consistency (Spearman's $R^2 > 0.99$, $P < 0.05$) in gene expression measurements across different sequencing platforms (Fig. S6). DEG analysis was then performed using control samples as a baseline. In the discovery cohort, 1,639, 1,802, and 2,407 DEGs were robustly identified in NAFL, Borderline, and NASH stages, respectively (Fig. 2a; Tables S2 to S4), which is independent of confounders, such as sequencing platform, RNA extraction method, library preparation protocol (Fig. S7). A core set of 358 consistently upregulated and 536 downregulated DEGs was shared across all three disease stages (Fig. 2b and c), including upregulated genes *IL32*, *AKR1B10*, and *FABP4* and downregulated genes *EGR1*, *CYP2C19,* and *VIL1* (Fig. 2a). Notably, Borderline exhibited fewer stage-specific DEGs than NAFL and NASH but shared more DEGs with NASH (Fig. 2b), indicating the transition from NAFL to Borderline as a critical tipping point.

Functional enrichment analysis revealed 23, 25, and 61 upregulated pathways and 34, 5, and 16 downregulated pathways in NAFL, Borderline, and NASH, respectively (Fig. S8; Tables S5 to S7). Consistent with the NAFL-to-Borderline transition observed in DEG analysis, enriched pathways distinguished NAFL from later stages. NAFL was featured by abnormal activation in ATP synthesis (Fig. 2d) and oxidative phosphorylation (Fig. S8b) and inhibition in organ development-related pathway (Fig. 2e; Fig. S8c). In contrast, Borderline and NASH shared a greater transcriptional similarity (Fig. 2d and e), driven by persistent mechanisms such as abnormal extracellular matrix organization (Fig. 2d; Fig. S8h), immune reactions (Fig. S8e and g), and xenobiotic catabolic processes (Fig. 2e; Fig. S8i). These findings collectively indicate that Borderline is a critical transcriptomic turning point in NAFLD progression, marked by significant shifts in gene expression and functional pathways that remain consistent in later stages. Notably, mitochondrion organization was activated across all disease stages, suggesting that these cellular processes may play a persistent role during the whole process (Fig. 2d).

## Turbulent hepatic microbiota involved in the hepatic microenvironment in NAFLD

Hepatic microbial profiles were obtained through a rigorous workflow, including microbial sequences mining, annotation, and decontamination (Fig. S2 and S3). The discovery cohort revealed 350 liver taxa from diverse domains: 296 bacteria, 46 fungi, seven viruses, and one archaeon. Importantly, these microbial profiles demonstrated consistent patterns across different rRNA library preparation methods (Spearman's $R^2 > 0.97$, $P < 0.05$) (Fig. S9) and sequencing platform (Fig. S10), confirming the robustness of our findings. The most abundant phyla were Proteobacteria and Actinobacteria in bacteria (11, 13), Basidiomycota and Ascomycota in fungi, and Artverviricota and Uroviricota in viruses (Fig. S11). Additionally, genera *Streptococcus*, *Acinetobacter*, and *Sphingomonas* (13) and species *Stenotrophomonas maltophilia* (10) and *Cutibacterium acnes* (41) were identified, consistent with previous findings. For instance, Proteobacteria was also identified as the most abundant phylum in NAFLD livers by 16S rRNA

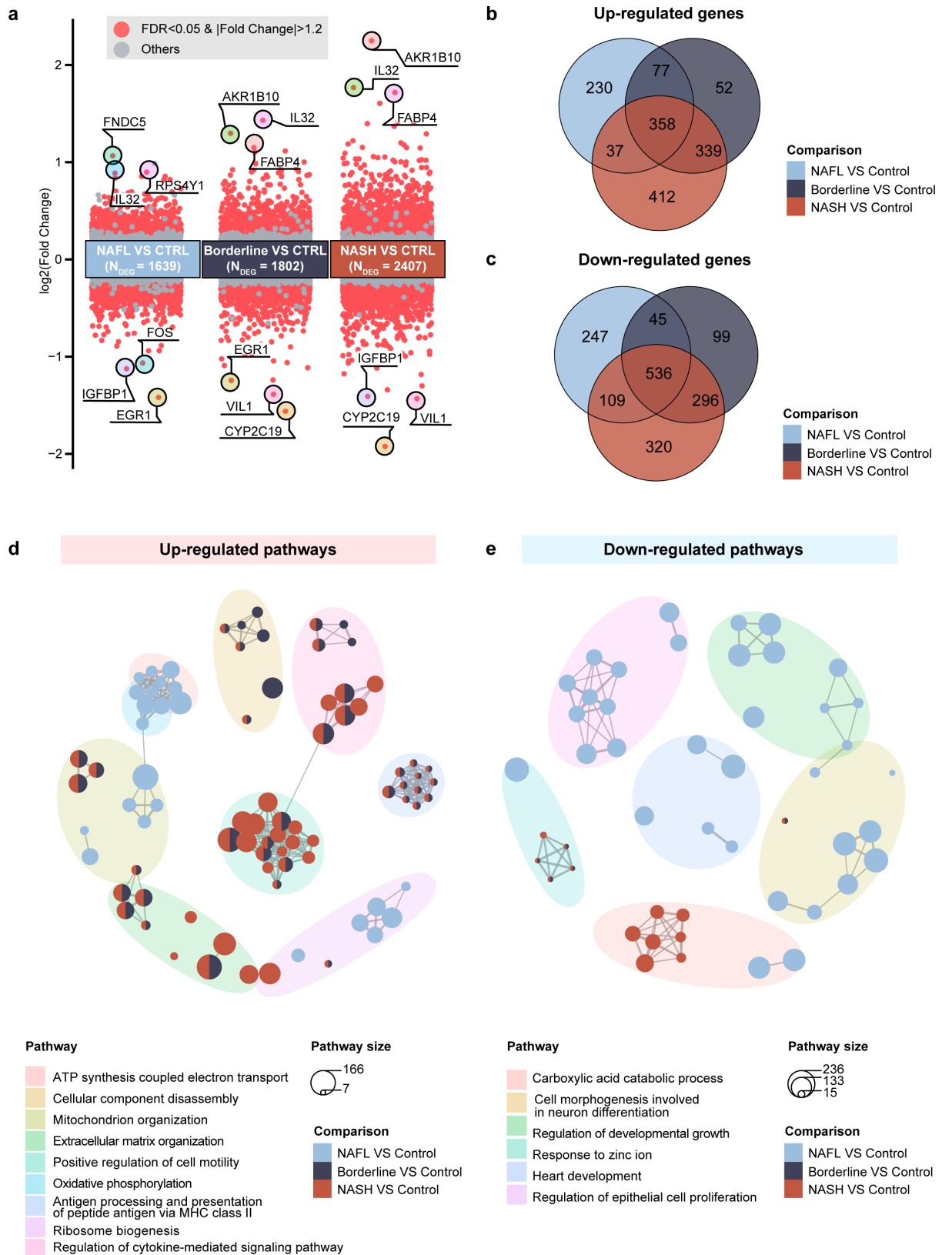

**FIG 2** Host gene characteristics associated with NAFLD progression. (a) Scatter plots illustrating the log2FC values of genes (*y*-axis) between three disease stages vs CTRL (control) (*x*-axis), red points represent DEGs whose FDR < 0.05 and absolute log2FC > 1.2. (b) Venn plot displaying the overlap of upregulated DEGs between three comparisons. (c) Venn plot displaying the overlap of downregulated DEGs between three comparisons. (d) Enrichment pathway plot showing

Fig 2 (Continued)

the upregulated pathways in the disease stage in each comparison with control, enriched by GSEA. Each dot represents a pathway; the dot size correlates with the number of genes in the pathway. Similar pathways were grouped into clusters, each shown in the same background color. The most significant pathway in a cluster was chosen as the representative of that cluster. (e) Enrichment pathway plot showing the downregulated pathways in the disease stage in each comparison with control, enriched by GSEA.

sequencing (11, 13). *S. maltophilia* has been specifically detected in cirrhotic livers through fluorescence *in situ* hybridization using probes targeting the 16S rRNA region (10).

To investigate the overall microbial characteristics throughout disease progression, we analyzed species-level diversity and dominant taxa dynamics. Alpha diversity revealed significantly higher richness and evenness in NAFL (Fig. 3a), while beta diversity revealed significant compositional shifts across stages (Fig. 3b). At the phylum level, disease progression featured declines in Actinobacteria and Fusobacteriota, and increases in Firmicutes and Ascomycota (Fig. S11). High-abundance taxa (above-average abundance) identified 37 species, 20 of which were shared across all four stages, collectively accounting for 77.54% of the total relative abundance (Fig. 3c). Among these, *C. acnes* and *Malassezia restricta* were the most abundant bacterium and fungus, respectively (Fig. 3d). *C. acnes* is known to maintain host homeostasis by producing short-chain fatty acids (41). Differentially abundant species identified 18, 12, and 17 significantly altered species in NAFL, Borderline, and NASH stages, respectively (Table S8). Eight bacteria consistently varied across all disease stages, including high-abundance taxa *Methyloversatilis* sp. RAC08 and *Ralstonia insidiosa*, which declined during disease progression (Fig. 3e).

Validation in a large external bulk RNA-seq data set confirmed the presence of 303 taxa (Table S9), with Proteobacteria and Actinobacteria remaining dominant phyla. Furthermore, single-cell RNA-seq analysis revealed microbial intracellular localization in 19,784 of 57,092 liver cells (Fig. S12a). In healthy livers, microbial taxa were significantly enriched in hepatocytes, natural killer (NK) cells, and T cells, while in NAFLD patients, they were predominantly found in macrophages, monocytes, and endothelial cells (Fig. S12b). Notably, 173 overlapping taxa were identified between the discovery cohort and the single-cell validation cohort (Table S10), with 162 also present in the bulk RNA-seq validation cohort, reinforcing the reliability of these findings.

## Disrupted hepatic microenvironment characterized by impaired host-microbiota interactions in NAFLD patients

Using Lasso regression, we identified host-microbiota interactions across stages (Fig. 4a), spanning all levels of microbial taxonomy to uncover a comprehensive understanding. Significant and stable interactions decreased from 5,537 in controls (Table S11) to 1,937 in NAFL (Table S12) and 1,455 in Borderline (Table S13), then increased to 2,933 in NASH (Table S14). Involved genes (Fig. 4b) and taxa (Fig. 4c) followed a similar overall decrease. An overall decline in both positive and negative interaction strength was observed as the disease progressed. Interestingly, there was a moderate increase in interaction strength from NAFL to the Borderline stage (Fig. 4e), suggesting a certain instability in the interaction network at this phase. However, as the disease advances to the NASH stage, a significant reduction in interaction strength was observed (FDR < 0.0001 with Kruskal-Wallis rank sum tests for both positive and negative interactions), indicating a progressive weakening of the communication network in the advanced stage. After controlling for factors including age, sex, sequencing instrument, NAS remained significantly associated with interaction strength, confirming that the observed relationship is independent of these potential confounders (Table S15) and showed consistent in the external cohort (Fig. S13). While no interactions were identical across all four stages, partial overlaps were observed in the disease stages. For example, 487 interactions were shared in NAFL and NASH, and the interaction "FGF21-Micromonospora" persisted across NAFL, Borderline, and NASH (Fig. 4d). These shared patterns suggest an early shift in

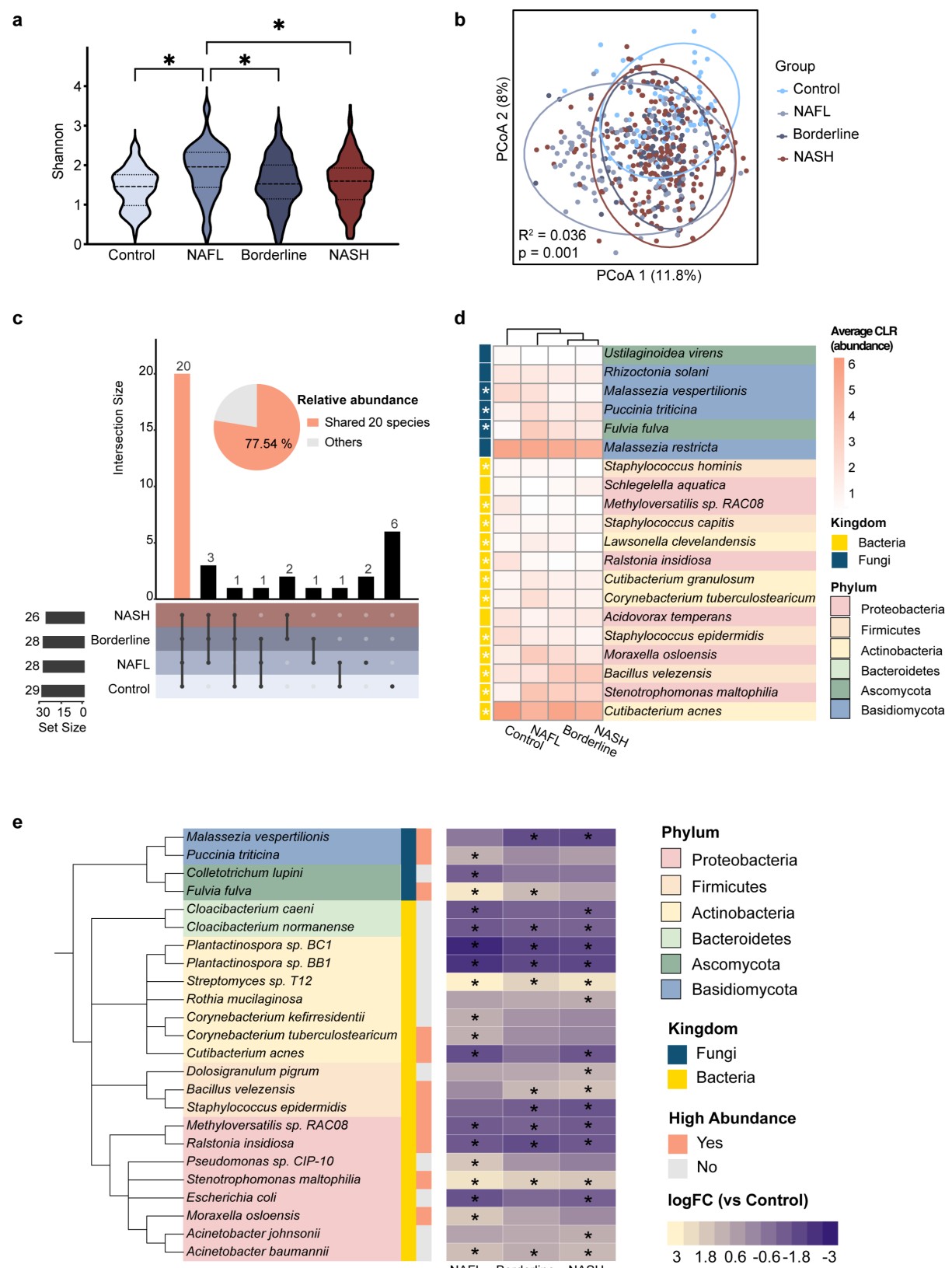

**FIG 3** Hepatic microbiota characteristics associated with NAFLD progression. (a) Shannon diversity of hepatic microbiota in different stages at species level. *P*-values were assessed via the Kruskal-Wallis test, with FDR values evaluated using *post hoc* Bonferroni correction (*FDR < 0.05). (b) Principal coordinate plot illustrating the differences across stages based on Aitchison distance of batch effect-adjusted hepatic microbiota abundances at species level (*P* = 0.001). (Continued on next page)

Fig 3 (Continued)

The *P*-value was calculated using permutational multivariate analysis of variance. The colors of the dots correspond to the different sample stages. (c) Upset plot depicting the intersects of high-abundant taxa in different stages. High-abundant taxa were defined as species whose average abundance exceeded the geometric mean of samples in the same stage. Twenty high-abundant taxa were shared in four stages, and the corresponding bar was highlighted in orange. The pie chart shows the relative abundance coverage of these 20 shared high-abundant taxa. (d) Heatmap illustrating the CLR-transformed abundance of these 20 shared high-abundant taxa in four stages. Annotation bar on the left indicates the kingdom to which each taxon belongs. Taxa marked with an asterisk mean a significant difference was observed in abundance between stages (FDR < 0.05), as determined using the Kruskal-Wallis test and corrected with *post hoc* Bonferroni method. The background color of the taxon name corresponds to the specific phylum it belongs to. (e) Heatmap with phylogenetic tree demonstrating differentially abundant taxa in each disease stage compared to the control stage. Differentially abundant taxa were defined as taxa whose FDR < 0.05 and absolute log2FC > 1. The background color of the taxon name corresponds to the belonging phylum. The left column of the annotation bar categorizes each taxon by kingdom, and the right column denotes whether the taxon was previously identified as highly abundant. Cells marked with an asterisk indicate that the taxon was identified as differentially abundant in the corresponding comparison. Upward arrows indicate a consistently increasing trend with disease progression, while downward arrows indicate the opposite trend.

interactions from control to NAFL, followed by relative stability during later stages (Fig. 4d).

Notably, dominant interactive taxa shifted between healthy and diseased states. In controls, *Escherichia coli*, *Actinomyces naeslundii,* and *Pseudomonas oleovorans* were prominent, but they participated in fewer interactions in disease stages (Fig. 4f). In contrast, these interaction patterns exhibited similarity among three disease stages. For example, Sphingomonadales and Sphingomonadaceae were the most interactive taxa in NAFL and Borderline, and *R. insidiosa* interacted with the most genes in NAFL and NASH (Fig. 4f). This underscores the potential of host-microbial interaction networks as early indicators of disease, as it provided an earlier shift during the control-to-NAFL than the NAFL-to-Borderline from host transcriptomic data.

## Hepatic microbiota was associated with host mitophagy and immune response in NAFLD

SparseCCA identifies interaction clusters where host genes and microbiota were most relevant. Pathway enrichment analysis revealed functional roles of these interactions across stages. We identified 145, 23, and 101 pathways in the control, NAFL, and Borderline stages, respectively, whereas no significant pathways were identified in NASH (Fig. 5a; Table S16). This suggests interactions in NASH were too weak to form cohesive biological functions, a trend also observed in the external cohort where fewer pathways were enriched in more advanced stages (Table S17). Stage-specific pathways indicated unique functions of interactions in their respective microenvironment (Fig. 5a). In control, interactions were linked to autophagy and unsaturated fatty acids biosynthesis processes; in NAFL, they were associated with ferroptosis process (Fig. 5b), and in Borderline, they may influence mTOR signaling, PI3K-Akt pathway, and antigen processing and presentation (Fig. 5b).

Stage-shared pathways may represent core biological signatures. Thus, we focused on the five pathways shared between NAFL and Borderline, with the mitophagy pathway being the most significant (Fig. 5b). Notably, previously identified active taxa (Fig. 4f), such as Bacillales, *R. insidiosa,* and Sphingomonadaceae, were closely associated with the mitophagy process in both NAFL and Borderline (Fig. 5c and d). To confirm mitophagy perturbation, we analyzed related gene expression. Interestingly, while these genes showed progressive upregulation from control through NAFL to Borderline stages, their expression levels plateaued in NASH instead of continuing to increase (Fig. S12a through d). This pattern coincided with a significant reduction in Sphingomonadaceae abundance in NASH (Fig. S15), suggesting that the loss of this microbial population may attenuate the mitophagy activation response observed in earlier disease stages. Intriguingly, *STING*, a cytosolic sensor involved in xenophagy, also increased in disease stages (Fig. S14e). T cell receptor signaling pathway was focused on as it was also enriched in both NAFL and Borderline (Fig. 5b). Our results showed that microbial taxa such as Bacilli, Sphingomonadales, and *R. insidiosa* could interact with host genes

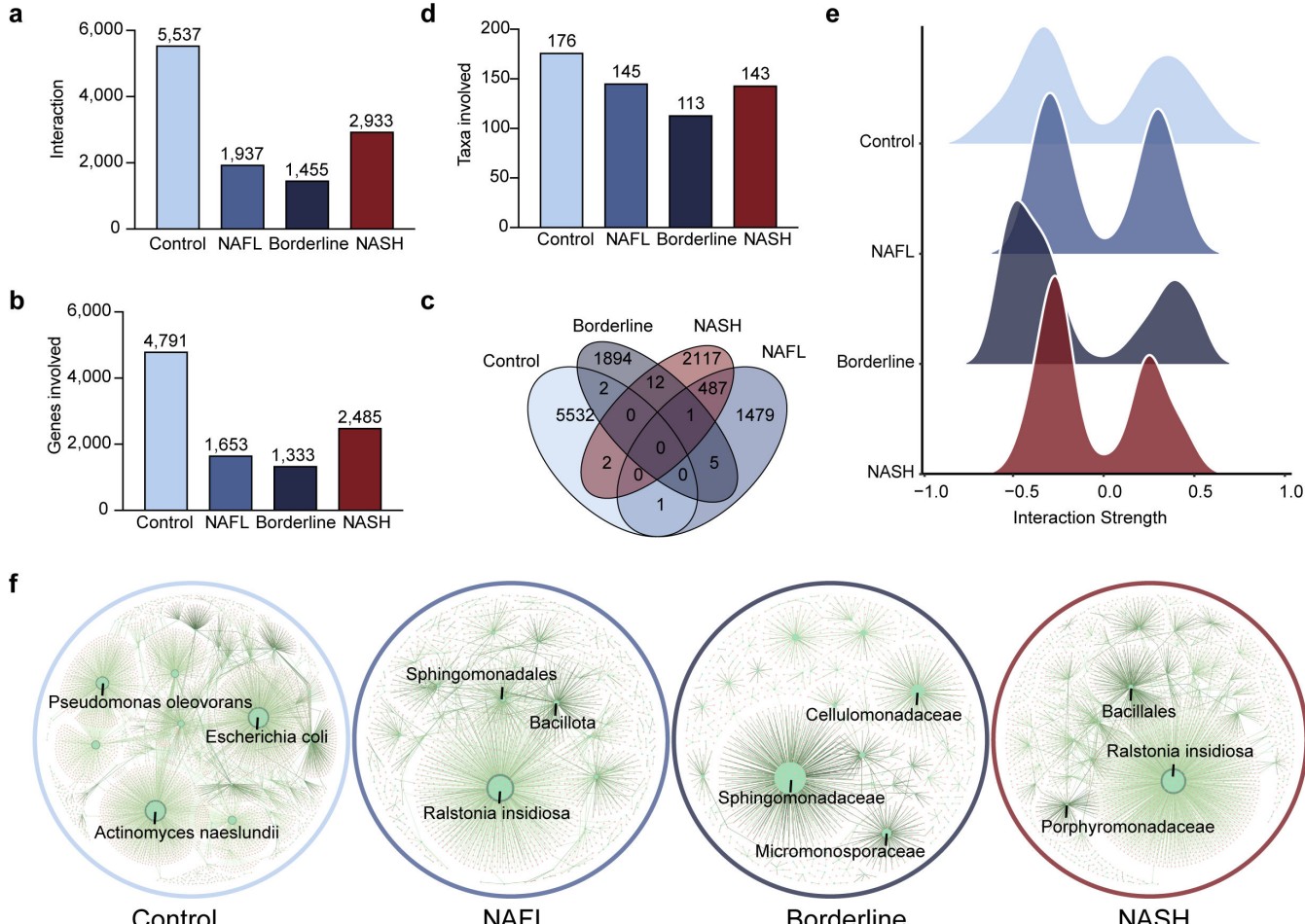

**FIG 4** Distinct characteristics of host gene-microbiota interactions across the NAFLD spectrum. (a) Bar plot displaying the total number of significant and stable host-microbiota interactions in each stage. (b) Bar plot indicating the number of genes involved in the interactions. (c) Bar plot indicating the number of taxa involved in the host-microbiota interactions. (d) Venn plot showing the overlap of shared interactions, which refers to the relationship between the same gene and taxon, in four stages. (e) Ridge plot showing interaction strength of the positive host-microbiota interactions (left) and negative host-microbiota interactions (right) in each stage, measured by Spearman correlation coefficients. The central line represents the median, while the outer lines represent the lower and upper quartiles. (f) Network plots illustrating the general host-microbiota interaction communities in each stage. Orange dots and green dots represent host genes and hepatic microbes involved in host-microbiota interactions, respectively. Node sizes are determined by node degree. The most gene-associated taxa were annotated.

including *PTPN11*, *PIK3CB*, *MAP2K7*, *HLA-DQB1*, *HLA-A,* and HLA-B. Expression of relevant genes was significantly upregulated (Fig. S14f through k), suggesting an immune-activated microenvironment.

Single-cell co-localization analysis revealed that cells expressing mitophagy-related genes indeed contained abundant relative microbes (Fig. S11b and c). These interactions were observed in macrophages, monocytes, T cells, and NK cells in control and mainly in hepatocytes and endothelial cells in NAFLD (Fig. S11b and c). In addition, T cell receptor signaling pathway-related interactions were confirmed to occurr in T cells in both control and NAFLD patients (Fig. S11d and e).

These results suggested that mitophagy, enriched in NAFL and Borderline but absent in NASH, may be critical for hepatic microbiota survival. In NASH, impaired related interactions could lead to reduced clearance of dysfunctional mitochondria and thus induce the release of reactive oxygen species (ROS) (Fig. 6). The subsequent ROS elevation can trigger inflammatory responses, induce genotoxic stress, and exacerbate

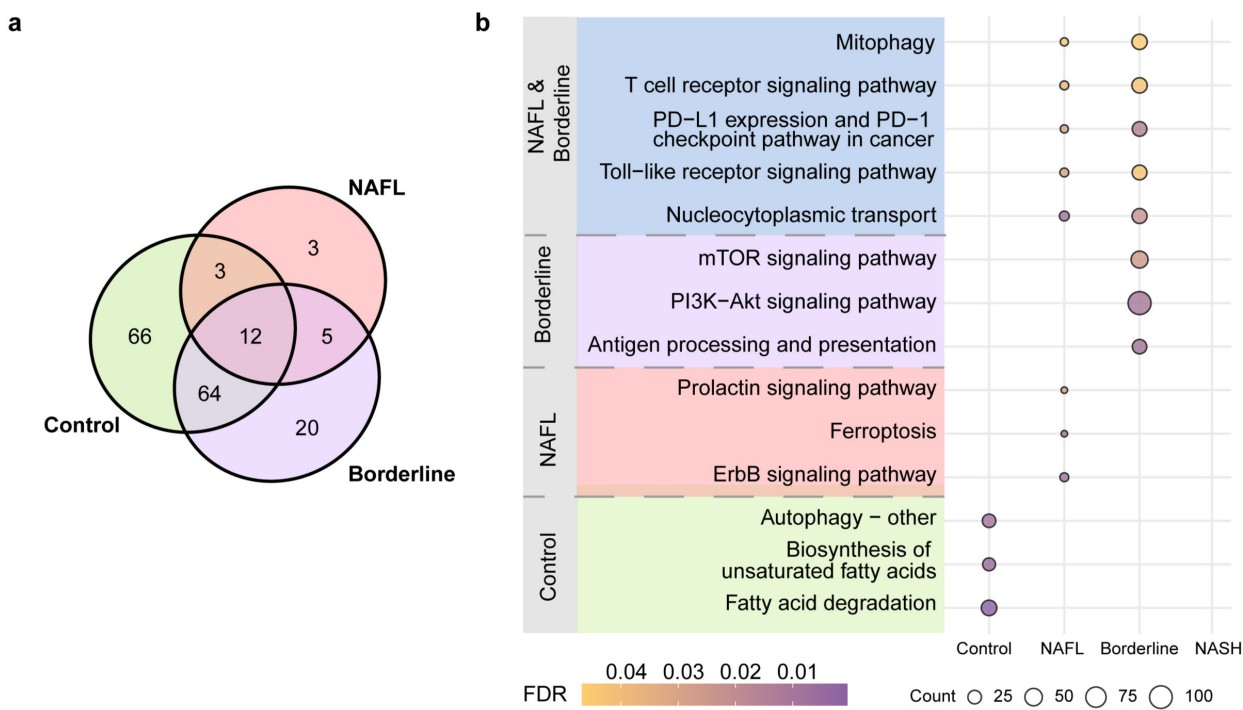

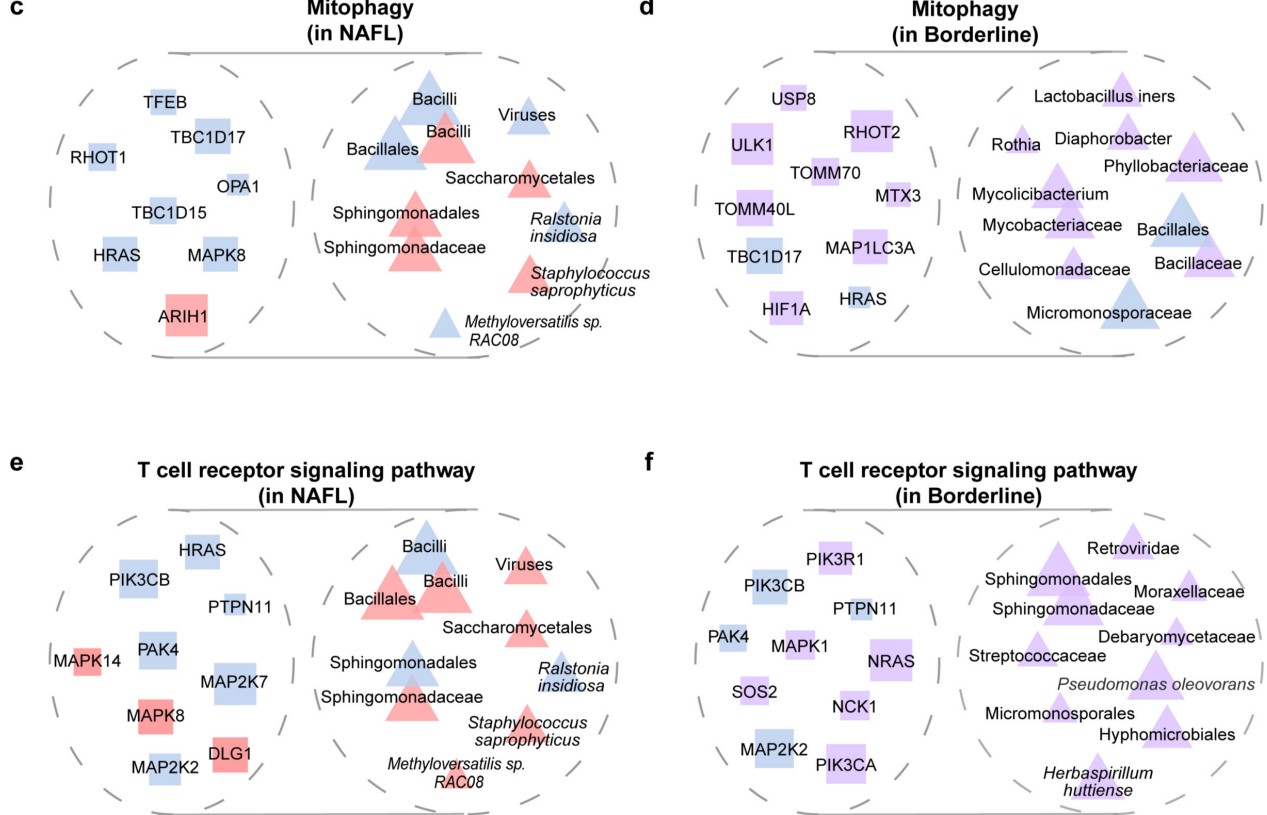

**FIG 5** Functional annotations of host-microbiota interactions. (a) Venn plot illustrating enriched pathways among genes correlated with specific hepatic microbes in control (green), NAFL (red), and Borderline (purple) (FDR < 0.05). No significant pathways were enriched for NASH. (b) Five pathways showing enrichment in both NAFL and Borderline are emphasized with a blue background color. Control-specific pathways are shown with a green background, (Continued on next page)

Fig 5 (Continued)

NAFL-specific pathways with a red background, and Borderline-specific pathways with a purple background. Dot sizes represent the number of genes in the pathways, and dot colors denote their significance after *post hoc* Bonferroni correction. (c, d) Association between hepatic microbiota and host genes within the mitophagy pathway, shared in both NAFL (c) and Borderline cohorts (d). The size of squares and triangles represents the absolute value of SparseCCA coefficients of genes and microbes, respectively. Genes and taxa that appeared in both NAFL and Borderline are shown in blue. Overlapping triangles indicate microbial taxa from the same taxonomic clade. (e, f) Association between hepatic microbiota and host genes within the shared T cell receptor signaling pathway in NAFL (e) and Borderline cohorts (f).

cell injury. Moreover, diminished mitophagy in NASH could amplify xenophagy, enabling more hepatic microbes to be degraded by the host (Fig. 6).

## DISCUSSION

This proof-of-concept study examines the full spectrum of NAFLD progression, from control, NAFL, the transient stage Borderline, to the advanced stage NASH. By incorporating multiple cohorts from diverse geographical regions and employing rigorous microbial RNA fragment analysis, the study provides general and robust evidence of the role of hepatic microbiota in influencing host mitochondrial function and disease progression.

A well-known source of hepatic microbiotas is the gut, as disruptions to the intestinal epithelial and gut vascular barriers are early events in NAFLD pathogenesis (6, 42). This gut-liver axis facilitates a complex bidirectional crosstalk, where microbial metabolites like secondary bile acids enter the liver via portal circulation (8), while liver secretions influence gut ecology through biliary excretion (9). For example, our previous work demonstrated how gut microbiota contributes to NAFLD by elevating secondary bile acids that suppress hepatic FXR signaling (20). Notably, impaired permeability may permit translocation of microbial components and even intact bacteria, triggering hepatic inflammation through pattern recognition receptors (43). However, unlike fecal microbiota dominated by Bacteroidetes and Firmicutes (44), hepatic microbiota show distinct enrichment of phyla Proteobacteria and Actinobacteria (11, 13), suggesting a selective filtering process during hepatic translocation or niche-specific adaptation. This distinction underscores the necessity of directly studying hepatic microbes to accurately capture the unique hepatic microenvironment. Specifically, our results and another recent study using 16S rRNA sequencing both identified the species *Corynebacterium tuberculostearicum* within the hepatic microbiota (45), highlighting a marked divergence between hepatic and fecal microbiome compositions. Such liver-specific enrichment strongly suggests a specialized role of *C. tuberculostearicum* within the hepatic microenvironment, potentially linked to abnormal fatty acid oxidative phosphorylation activity (46). This observation expands the recognized role of this bacterium beyond its traditional association with skin and respiratory diseases (45, 47). Furthermore, our finding that over 50% of taxa are shared between bulk and single-cell data sets, along with their presence in immune cells such as macrophages, monocytes, and T cells, highlights the robustness of our results and suggests a potential functional relationship between hepatic microbes and liver-resident immune cells. Microbe-carrying cells have been shown to enhance host-cell survival by increasing resistance to fluid shear stress during circulation (48). However, the precise origins of hepatic microbiota require further exploration, particularly using advanced imaging techniques like fluorescent D-amino acid-based metabolic labeling (49). On the other hand, we found that under healthy conditions, hepatic microbes primarily resided within monocytes and T cells, suggesting that immune cells act as the first line of defense by sequestering microbes and preventing their spread (50). During NAFLD progression, microbes shift to endothelial cells, potentially exacerbating inflammation and liver injury. This shift suggests that the compromised defense mechanisms facilitate microbes to escape immune cell sequestration and expand within liver endothelial cells, disrupting vascular integrity

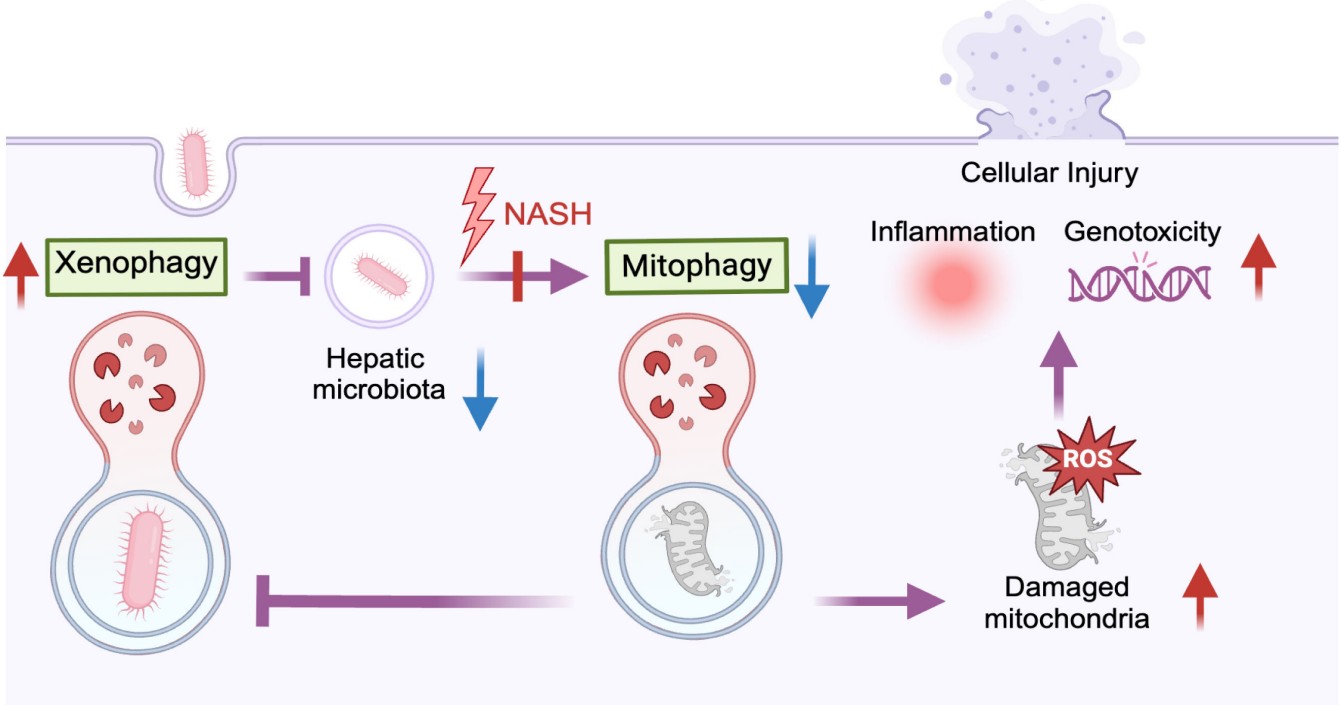

**FIG 6** Potential mechanisms for interaction between hepatic microbiota and host in affecting NAFLD disease progression. Hepatic microbes assist in enhancing the host mitophagy process, which further inhibits the xenophagy process to ensure their survival in cells. Such interaction is diminished in NASH patients. Impaired clearance of damaged mitochondria could lead to toxic chemicals like ROS releasing, which could further result in genotoxicity, inflammation, and cellular damage.

and promoting further hepatic injury (6). Further research is needed to clarify their mechanisms in NAFLD progression.

Our comprehensive analysis identifies the Borderline stage as a critical transition point in NAFLD progression (27). The transcriptomic signature of the Borderline stage demonstrates a fundamental biological transition in NAFLD progression, characterized by the coexistence of compensatory metabolic adaptation such as oxidative phosphorylation and emerging pathological remodeling such as extracellular matrix reorganization. This unique combination creates biological instability in this phase. Importantly, this transitional period also exhibits unique host-microbe dynamics, characterized by remarkable interaction plasticity. While overall host-microbe network strength shows a declining trend, we observe pronounced fluctuation in interaction intensities during Borderline compared to other stages, indicating a last-ditch effort to maintain homeostasis before the system collapses into the rigid, dysfunctional state characteristic of NASH (4). As the disease progresses to NASH, host-microbiota interactions become weakest, reflecting adaptations to a deteriorating hepatic environment and contributing more to the pathologic state than to homeostasis. For instance, Porhyromonadaceae, a taxon linked to liver inflammation (51, 52), emerged as a dominant player in NASH, replacing earlier active taxa such as Sphingomonadaceae and Sphingomonadales. Interestingly, microbial shifts reveal that the control-to-NAFL transition already exhibits early changes in the hepatic microenvironment. Notably, *E. coli*, *A. naeslundii*, and *P. oleovorans* were the most associated taxa in control and were replaced by Sphingomonadaceae in NAFL and Borderline. These shifts from control to NAFL in microbial activity suggest that the liver microbiome interaction network could detect microenvironment changes earlier than transcriptomic shifts, offering a potential window for early diagnosis, risk stratification, and targeted interventions in disease progression (14).

The transition to NASH involves profound microenvironmental disruption, marked by distinct transcriptomic changes and sustained activation of pro-inflammatory pathways. Our results showed upregulated chemokine production process, which could create an immunological environment that fundamentally alters microbial survival and function. This inflammatory pressure could therefore select for pathogenic bacterial species while depleting beneficial commensals like Sphingomonadaceae. A key manifestation of this breakdown is the dysregulation of mitophagy. While the importance of mitophagy in NAFLD has been well established (10, 53–55), the mechanisms underlying its impairment remain incompletely understood. Our study advances understanding by identifying Sphingomonadaceae as a crucial microbial modulator of mitophagy (56–59). Mechanistically, this regulation likely occurs through sphingolipid-mediated signaling (60), as supported by single-cell resolution data showing preferential enrichment of these host-microbe interactions in T cell populations (61). This symbiotic relationship exhibits evolutionary optimization: while microbial sphingolipids support host immune cell function by maintaining mitochondrial homeostasis (61, 62), the microbes themselves benefit through mitophagy-mediated suppression of xenophagy for niche persistence (59). The collapse of this delicate balance in NASH carries significant pathological consequences. With Sphingomonadaceae populations declining and their sphingolipid signaling diminished, the resulting mitophagy impairment leads to the accumulation of dysfunctional mitochondria and excessive ROS production (63). This oxidative burden, compounded by additional ROS sources including NADPH oxidase (64), peroxisomes (65), and the cytochrome P450 enzyme system (66), contributes to hepatocyte injury through genotoxic stress and chronic inflammation (67, 68). Clinically, these findings suggest that the Sphingomonadaceae represents a potential therapeutic target that could be targeted as a more natural mitophagy trigger (69). By preserving or restoring these specific microbial interactions, potentially through targeted probiotics, fecal microbiota transplantation, or dietary modulation of sphingolipid-producing bacteria such as Sphingomonadaceae, we may achieve more physiologically balanced regulation of mitochondrial quality control compared to direct pharmacological mitophagy induction. Future experimental studies, including gnotobiotic animal models and microbial supplementation trials, will be essential to validate this therapeutic potential and optimize strategies for clinical translation.

A common challenge when analyzing sequencing data is to obtain high repeatable results, with variabilities between different experimental batches, which is also known as batch effect (70, 71). To identify core and fundamental biological signals driving pathogenesis that persist across diverse populations, we integrated multiple data sets and applied rigorous batch effect correction (72). With this approach, we were able to uncover consistent mechanisms aligned with prior studies (11, 13, 56–59, 73–76) and external cohorts (33, 34). Furthermore, the host-microbiota interaction network remains significantly weak after controlling for potential confounding factors, confirming the robustness of our analytical strategy. However, inherent limitations remain. First, technical factors including RNA extraction methods and library preparation protocols may influence microbial detection sensitivity in tissue transcriptomes, as bacterial mRNA lacks polyA tails and microbial biomass is typically low. Second, while our k-mer-based taxonomic profiling provides reliable community composition data, we recognize that direct functional analysis, including gene annotation and open reading frame characterization, would offer more direct insights into microbial metabolic potential. However, such analysis would require substantially deeper sequencing depth to enable reliable metagenomic assembly. Third, factors such as variations in age, diet, and lifestyle across cohorts may still influence microbial composition and are challenging to fully disentangle (71). Fourth, while control samples from bariatric surgery patients provide relevant metabolic comparisons, they limit insights into early susceptibility to NAFLD. Future studies should integrate optimized methods for capturing microbial RNA, such as pure RNA extraction methods and random-primed library preparation methods, along with

samples from healthy controls and deeper sequencing, to better understand the initial mechanisms of the disease.

In summary, this study has comprehensively demonstrated the microenvironment across the whole progression of NAFLD, using an integrated analysis across multiple cohorts. Our results suggested that disruption of host-microbiota interactions, especially those who are vital for mitophagy, may promote disease progression. In addition, shifts in host-microbe interactions could serve as an early indicator of microenvironmental changes in NAFL, enabling proactive interventions before the disease progresses to the critical Borderline stage. Collectively, our results shed light on a fresh perspective in the multifaceted mechanisms driving NAFLD progression, providing insights for potential therapy and intervention strategies targeting hepatic microbes.

## ACKNOWLEDGMENTS

This work was supported by the National Natural Science Foundation of China (32470098 to N.J., 92251307 to R.Z., 82170542 to R.Z.) and the National Key Research and Development Program of China (2021YFF0703700/2021YFF0703702 to R.Z.). The funders had no role in study design, data collection and analysis, decision to publish, or preparation of the manuscript.

N.J., L.Z., and R.Z. conceived and designed the project. W.Y. and W.G. drafted the manuscript. YY, W.L., W.C., X.Z., R.Z., L.Z., and N.J. revised the manuscript. All authors read and approved the final manuscript.

## AUTHOR AFFILIATIONS

[1]Putuo People's Hospital, School of Life Sciences and Technology, Tongji University, Shanghai, China
[2]Institutes of Biomedical Sciences, School of Life Sciences, Inner Mongolia University, Hohhot, China
[3]State Key Laboratory of Genetic Engineering, Fudan Microbiome Center, School of Life Sciences, Fudan University, Shanghai, China

## AUTHOR ORCIDs

Wenjing Yin http://orcid.org/0009-0007-1262-3933
Ruixin Zhu http://orcid.org/0000-0002-5070-6453
Lixin Zhu http://orcid.org/0000-0001-7904-1769
Na Jiao http://orcid.org/0000-0003-3976-6313

## FUNDING

| Funder | Grant(s) | Author(s) |
| --- | --- | --- |
| National Natural Science Foundation of China | 32470098 | Na Jiao |
| National Natural Science Foundation of China | 92251307, 82170542 | Ruixin Zhu |
| National Key Research and Development Program of China | 2021YFF0703700, 2021YFF0703702 | Ruixin Zhu |

## AUTHOR CONTRIBUTIONS

Wenjing Yin, Data curation, Formal analysis, Investigation, Methodology, Resources, Software, Validation, Visualization, Writing – original draft, Writing – review and editing | Wenxing Gao, Formal analysis, Investigation, Methodology, Visualization, Writing – review and editing | Yuwei Yang, Investigation, Writing – review and editing | Weili Lin, Investigation, Writing – review and editing | Wanning Chen, Writing – review and editing | Xinyue Zhu, Writing – review and editing | Ruixin Zhu, Conceptualization, Funding acquisition, Investigation, Methodology, Project administration, Supervision, Writing – review and editing | Lixin Zhu, Conceptualization, Funding acquisition, Investigation,

Project administration, Supervision, Writing – review and editing | Na Jiao, Conceptualization, Funding acquisition, Investigation, Methodology, Project administration, Supervision, Writing – review and editing

## DATA AVAILABILITY

All of the processed data in this study have been uploaded in the National Omics Data Encyclopedia under accession number OEZ00014588. The raw metagenomic data are available in the European Nucleotide Archive (https://www.ebi.ac.uk/ena/) under accession numbers PRJNA558102, PRJNA767535, PRJNA682622, PRJNA704861, and PRJNA523510. The data relevant to the study are uploaded as supplemental material. The code and scripts are available on GitHub (https://github.com/tjcadd2020/NAFLD-Host-Microbe-Interaction).

## ADDITIONAL FILES

The following material is available online.

### Supplemental Material

**Supplemental material (Spectrum00100-25-s0001.pdf).** Fig. S1 to S15 and legends for Tables S1 to S17.
**Supplemental tables (Spectrum00100-25-s0002.pdf).** Tables S1 to S17.

### Open Peer Review

**PEER REVIEW HISTORY (review-history.pdf).** An accounting of the reviewer comments and feedback.

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
