## [Reviewer comments · Microbiology Spectrum]

Microbiology Spectrum

Disrupted Host-Microbiota Crosstalk Promotes Nonalcoholic Fatty Liver Disease Progression by Impaired Mitophagy

Wenjing Yin, Wenxing Gao, Yuwei Yang, Weili Lin, Wanning Chen, Xinyue Zhu, Ruixin Zhu, Lixin Zhu, and Na Jiao

Corresponding Author(s): Na Jiao, Fudan University

Review Timeline:

Submission Date:	January 9, 2025
Editorial Decision:	March 8, 2025
Revision Received:	April 2, 2025
Accepted:	April 5, 2025

Editor: Benjamin Liu

Reviewer(s): The reviewers have opted to remain anonymous.

Transaction Report:

DOI: <https://doi.org/10.1128/spectrum.00100-25>

Re: Spectrum00100-25 (Disrupted Host-Microbiota Crosstalk Promotes Nonalcoholic Fatty Liver Disease Progression by Impaired Mitophagy)

Dear Prof. Na Jiao:

Thank you for the privilege of reviewing your work. Below you will find my comments, instructions from the Spectrum editorial office, and the reviewer comments.

Editor's comments:

1. In NASH, impaired related interactions could lead to reduced clearance of dysfunctional mitochondria, increasing reactive oxygen species (ROS). Disruption to the equilibrium around mitophagy leads to the accumulation of damaged mitochondria and increased production of ROS. In Figure 6, ROS is illustrated after severe mitochondria damage. But the authors failed to mention there are different sources of ROS generation in liver exposed to toxic and pathogenic processes but without differentiation and detection in the authors' cohorts (PMID: 39457713). Besides triggering inflammation, ROS can lead to genotoxicity (PMID: 39457713). For example, direct and indirect DNA damage pathways can lead to double-strand DNA breaks, single-strand breaks, and mismatches, thereby altering DNA replication, stalling transcription, and PARP activation (PMID: 39457713). ROS can cause necrosis, cellular stress, and apoptosis. DNA damage can also cause apoptosis (PMID: 39457713). These points should be added to discussion. More references should be cited, with this one (PMID: 39457713) as an example (citing is optional).

2. In Fig. 3, the authors identified *Corynebacterium* spp as part of the hepatic microbiota associated with NAFLD progression. However, *Corynebacterium* spp is a normal cutaneous flora. The authors should discuss the physiological roles of *Corynebacterium* spp in humans and significance of detection of these organisms in their experiments. More references should be cited, with this one (PMID: 34406882) as an example (citing is optional).

3. What are the inclusion and exclusion criteria in this study? Did the authors exclude viral hepatitis?

Please return the manuscript within 30 days; if you cannot complete the modification within this time period, please contact me. If you do not wish to modify the manuscript and prefer to submit it to another journal, notify me immediately so that the manuscript may be formally withdrawn from consideration by Spectrum.

Revision Guidelines

Sincerely,
Benjamin Liu
Editor
Microbiology Spectrum

Reviewer #1 (Comments for the Author):

It is a detailed and well-structured scientific study on the role of disrupted host-microbiota interactions in NAFLD progression. This is a novel approach focusing on mitophagy disruption linked to microbiota changes in NAFLD, a relatively underexplored area in research. It is a planned multicohort design consisting of 570 samples across five independent cohorts. All stages of disease progression of NAFLD, Borderline and NASH were clearly defined, where this it depicts the Borderline stage is a transition stage and can potentially be reversed. The study uses advance innovative statistical models and bio-informative and machine learning tools like LASSO regression, SparseCCA for host-microbiota interaction analysis. Single-cell RNA-seq validation strengthens the findings at the cellular resolution.

Special comments that need to be consider:

The article is dense with technical terms. A more structured approach with short, impactful sentences could enhance readability. Clarity should be there in terminology and the author should first explain the methods in layman's terms before describing them in detail.

After the introduction of the associated research, one paragraph for the objectives statement should be there.

The transition between the change in microbial shifts and mitophagy could be made easier to read. The transition from gut microbiota to intrahepatic microbiota needs better clarification. Hence the authors can add more recent published articles in support of this.

The authors present detailed observations but do not offer insight interaction what is the underlying mechanism that explains these observations. The control was taken less (n= 72) where NAFL (n= 124), Borderline (n=143) and NASH (n= 231). The Borderline stage needed more focus as it came out to be the transition state.

The mechanistic explanation for interaction loss in NASH (vs. persistence in earlier stages) needs further expansion.

There were lacking evidence in the study, confirming that the microbial changes can cause mitophagy impairment.

Future studies should explore whether specific microbial interventions can enhance mitochondrial quality control in liver disease.

Reviewer #2 (Comments for the Author):

The title of the manuscript reflects the content of the manuscript.

Bacteria and some viruses lack a polyA tail in their mRNA. The polyA tail-based mRNA enrich method and pure RNA extraction (without DNA contamination)-based RNA-seq data should avoid bacterial reads. The RNA extraction methods: The TRIzol method gives pure RNA with less DNA contamination, but the column-based RNA extraction has more DNA contamination. Therefore, the RNA extraction method and the library preparation method are also considered in the cohort analysis.

HiSeq, MiSeq, and NovaSeq are all sequencing instruments made by Illumina, and they also have differences in gene enrichment. Therefore, this platform of sequencers also needs to be considered in the cohort analysis.

In the case of differential gene expression, proper normalization needs to be used and justified for RNA extraction method, the library preparation method and the sequencing platform. The total amount of data (reads in GB) generated, the number of gene coverage in each sample, and the number of total reads after trimming in each sample also affect the normalized gene count from different Biosample IDs.

The biopsy procedure and part of the liver samples also affect gene expression. How the control samples were collected, if autopsy samples, how many hours after death the samples were collected, also affects the gene expression.

How many bacteria are identified, how many genes of each bacterium are identified, what is the death coverage of each bacterial gene, how many reads per bacterial gene before and after normalization, what is the genome coverage of each bacterial gene, and the ORF nucleotide sequences details are not provided in this manuscript.

Any antibiotic-resistant ORF identified.

How many bacteria are identified, how many genes of each bacterium are identified, what is the death coverage of each bacterial gene, how many reads per bacterial gene before and after normalization, what is the genome coverage of each bacterial gene, and the ORF nucleotide sequences are co-localized in the single-cell sequencing is not provided.

After these analyses, one can conclude the findings of this manuscript.

Editor's comments:

1. In NASH, impaired related interactions could lead to reduced clearance of dysfunctional mitochondria, increasing reactive oxygen species (ROS). Disruption to the equilibrium around mitophagy leads to the accumulation of damaged mitochondria and increased production of ROS. In Figure 6, ROS is illustrated after severe mitochondria damage. But the authors failed to mention there are different sources of ROS generation in liver exposed to toxic and pathogenic processes but without differentiation and detection in the authors' cohorts (PMID: 39457713).

Answer: Thank you for your valuable comment. We fully agree with your insightful suggestion that, beyond mitochondria, several other cellular systems contribute to the overall ROS burden in the liver^{1,2}, such as NADPH oxidase complex³, peroxisomes⁴, cytochrome P450 enzyme system⁵ and immune cells⁶.

Nevertheless, our study specifically focuses on the ROS generated by dysregulated mitophagy for two main reasons. **First, mitochondria represent a principal and potent source of ROS⁷**, especially when their function is compromised⁸. Under normal conditions, mitochondria generate ROS as natural byproducts of oxidative phosphorylation, and efficient mitophagy is responsible for maintaining ROS at balanced levels⁹. When mitophagy is dysregulated in the context of NAFLD, damaged mitochondria accumulate, leading to a significant increase in ROS production^{8,9}. **Second, our results highlight a close association between ROS released from impaired mitophagy and hepatic microbe.** We propose that targeting these microbes might reduce mitochondrial ROS production, thereby potentially mitigating the inflammatory cascade and delaying NAFLD progression.

While our study concentrates on mitochondrial ROS due to its mechanistic relevance with hepatic microbes in NAFLD, we acknowledge the broader spectrum of ROS sources collectively shapes the hepatic microenvironment. Accordingly, we have revised our manuscript to incorporate a more comprehensive discussion of ROS origins (Line 499-501).

Besides triggering inflammation, ROS can lead to genotoxicity (PMID: 39457713). For example, direct and indirect DNA damage pathways can lead to double-strand DNA breaks, single-strand breaks, and mismatches, thereby altering DNA replication, stalling transcription, and PARP activation (PMID: 39457713). ROS can cause necrosis, cellular stress, and apoptosis. DNA damage can also cause apoptosis (PMID: 39457713). These points should be added to discussion. More references should be cited, with this one (PMID: 39457713) as an example (citing is optional).

Answer: We appreciate and fully agree that the downstream effects of ROS are multifaceted. Excessive ROS can directly damage cellular components, including lipids, proteins, and nucleic acids, thereby impairing cellular function and triggering cell death pathways such as apoptosis or necrosis¹⁰. In addition to causing direct cellular damage, ROS serve as signaling molecules that modulate various intracellular pathways⁸. They can activate transcription factors like NF- κ B and AP-1, leading to the production of pro-inflammatory cytokines, and they also affect the balance between pro-apoptotic and anti-apoptotic signals. This complex network of ROS-induced signaling contributes not

only to tissue injury but also to the chronic inflammatory milieu characteristic of NAFLD¹¹, where aberrant lipid accumulation and inflammation are key features.

In the revised manuscript, we have expanded our discussion to include these diverse roles and effects of ROS more comprehensively (Line 416-421,499-503). We have also updated Fig 6 to include these effects of ROS.

Fig 6. Potential mechanisms for interaction between hepatic microbiota and host in affecting NAFLD disease progression

Hepatic microbes assist in enhancing the host mitophagy process, which further inhibits the xenophagy process to ensure their survival in cells. Such interaction is diminished in NASH patients. Impaired clearance of damaged mitochondria could lead to the toxic chemicals like reactive oxygen species (ROS) releasing, which could further result in genotoxicity, inflammation and cellular damage.

2. In Fig. 3, the authors identified *Corynebacterium spp* as part of the hepatic microbiota associated with NAFLD progression. However, *Corynebacterium spp* is a normal cutaneous flora. The authors should discuss the physiological roles of *Corynebacterium spp* in humans and significance of detection of these organisms in their experiments. More references should be cited, with this one (PMID: 34406882) as an example (citing is optional).

Answer: We sincerely thank you for highlighting the intriguing observation regarding the presence of *Corynebacterium spp.* in the hepatic microbiota composition. Traditionally recognized as a constituent of the skin microbiome, this genus has recently been detected in liver samples using 16S rRNA sequencing¹². This finding suggests that *Corynebacterium spp.* **may extend ecological roles extending beyond cutaneous environment to other tissues, including the liver**^{12,13}. This observation aligns with our results and underscores the importance of investigating hepatic microbiota, as it could uncover novel functions of these well-characterized microbes.

Although the precise origins and translocation mechanisms of *Corynebacterium spp.* into the liver remain unclear, their potential contributions to liver physiology and pathology need further investigation. In traditional dermatological conditions, such as pitted keratolysis and erythrasma, *Corynebacterium spp.* have been implicated in modulating inflammatory responses¹⁴ and regulating lipid metabolism¹⁵. These processes are particularly relevant

to liver homeostasis, yet the involvement of *Corynebacterium spp.* in NAFLD pathogenesis remains underexplored. Notably, prior research has demonstrated that a specific species, *Corynebacterium tuberculostearicum*, is significantly associated with increased fatty acid oxidative phosphorylation activity¹⁵. In our study, **this species was observed a significant increase in NAFL stage compared to healthy samples (Log2FC = 1.184, FDR < 0.05)**. Such findings suggest a potential connection between this microbial species and NAFLD pathogenesis.

In summary, the detection of *Corynebacterium spp.* not only broadens our understanding of its role in human health but also provides new insights into mechanism of NAFLD pathogenesis. Further exploration of host-associated microbes, such as this species, could unveil critical mechanisms underlying disease progression and potentially guide the development of novel therapeutic interventions. These points have been incorporated into the revised manuscript, along with additional references, to provide a more comprehensive context (Line 440-447).

3. What are the inclusion and exclusion criteria in this study? Did the authors exclude viral hepatitis?

Answer: We thank the editor for this comment, which has helped us improve the clarity of our manuscript. For the inclusion criteria of our research, we sourced public datasets using keywords such as "RNA-seq," "transcriptomic," and "liver biopsies", focusing specifically on cohorts **providing both transcriptomic profiles and clearly defined sample group information**. To ensure rigor and consistency in cohort selection, we implemented a comprehensive workflow defining clear exclusion criteria, as illustrated in Figure S1. Specifically, we excluded study PRJNA512027 **due to the lack of a clear clinical diagnostic standard**, which precluded reliable classification of samples into NAFL, Borderline, and NASH groups. Additionally, dataset PRJNA542148 was excluded because it **failed to meet our basic quality control standards**, with an average qualified read count below 20 million. This threshold was chosen based on recommendations from the ENCODE Consortium, which suggests a minimum sequencing depth for reliable transcriptome profiling¹⁶. Consequently, our discovery dataset comprises the remaining five cohorts, totaling 570 samples.

We further emphasize that our study strictly adheres to the established clinical definition of NAFLD, **explicitly excluding samples from individuals with viral hepatitis or significant alcohol consumption**¹⁷. To rigorously ensure compliance with this criterion, we manually reviewed the original publications for each cohort, confirming that appropriate exclusion criteria regarding viral hepatitis and alcohol intake were explicitly stated.

A detailed description of these inclusion and exclusion criteria has now been clearly added to the Methods section of our revised manuscript (Line 125-132).

Figure S1. Discovery cohort collection and sample filtering criteria

Reviewer #1 (Comments for the Author):

It is a detailed and well-structured scientific study on the role of disrupted host-microbiota interactions in NAFLD progression. This is a novel approach focusing on mitophagy disruption linked to microbiota changes in NAFLD, a relatively underexplored area in research. It is a planned multicohort design consisting of 570 samples across five independent cohorts. All stages of disease progression of NAFLD, Borderline and NASH were clearly defined, where this it depicts the Borderline stage is a transition stage and can potentially be reversed. The study uses advance innovative statistical models and bio-informative and machine learning tools like LASSO regression, SparseCCA for host-microbiota interaction analysis. Single-cell RNA-seq validation strengthens the findings at the cellular resolution.

Answer: We sincerely appreciate the reviewer's thoughtful evaluation and positive recognition of our study's contributions.

Special comments that need to be consider:

The article is dense with technical terms. A more structured approach with short, impactful sentences could enhance readability. Clarity should be there in terminology and the author should first explain the methods in layman's terms before describing them in detail.

Answer: Thank you for this valuable suggestion. In response, we have thoroughly restructured the manuscript to improve clarity and readability. All methods sections now begin with a concise, lay-person summary that explains the purpose and overall approach in plain language before presenting technical details. Technical terminology is defined at first use, and long, complex sentences have been replaced with shorter, more impactful statements (Line 101-103, 115-121, 153-157, 161-176, 199-210, 214-232, 238-265).

After the introduction of the associated research, one paragraph for the objectives statement should be there.

Answer: We sincerely appreciate this constructive suggestion. In response, we have added a dedicated objectives paragraph at the conclusion of the introduction to clearly articulate our study goals (Line 106-112). Specifically, this new paragraph states our three primary aims: (1) to systematically profile the composition of hepatic microbiota in NAFLD progression, (2) to characterize host-microbe interaction patterns through multi-omics integration, and (3) to evaluate the potential mechanistic contributions of these interactions to disease pathogenesis. We believe this addition provides readers with a clearer roadmap of our study's purpose and scientific value before delving into methodological details.

The transition between the change in microbial shifts and mitophagy could be made easier to read.

Answer: We sincerely appreciate the reviewer's suggestion for helping us make our manuscript more logically structured and reader-friendly. The revised discussion now follows a clearer progression: beginning with an overview of hepatic microbiota (Line 429-460), then examining microenvironmental changes during NAFLD progression (Line 461-481). Based on this, we specifically highlight mitophagy dysregulation as a key manifestation of such deteriorating hepatic microenvironment in NASH (Line 482-509). This organization allows readers to first understand the broader context of

microenvironmental breakdown before focusing on the specific mechanism of mitophagy impairment. By positioning mitophagy disruption within this logical framework, we believe the manuscript now more effectively communicates both the overall pathological changes and their specific mechanistic consequences.

The transition from gut microbiota to intrahepatic microbiota needs better clarification. Hence the authors can add more recent published articles in support of this.

Answer: Thank you for this insightful suggestion. We fully agree that establishing this connection is crucial for contextualizing our study. In response, we have strengthened both the introduction and discussion sections with references (Line 85–97, 429-447) that specifically address microbial translocation along the gut-liver axis.

Gut microbiota serves as the primary source of intrahepatic microbes, with microbial products and even intact bacteria potentially translocating to the liver through impaired gut barriers or via portal circulation^{18,19}. In addition, our findings and others' work highlight that intrahepatic microbiota develop distinct compositional profiles compared to gut^{20,21}, a phenomenon likely influenced by hepatic immune surveillance and local microenvironment selection pressures. These differences underscore why direct investigation of liver-resident microbes is essential for understanding their unique roles in liver pathophysiology.

We believe these clarifications, supported by the newly added literature, significantly enhance the logical flow and scientific rigor of our manuscript. The improved transitions better guide readers from the well-established gut microbiota concepts to our novel focus on intrahepatic microbial communities and their potential clinical significance in NAFLD.

The authors present detailed observations but do not offer insight into the underlying mechanism that explains these observations.

Answer: We sincerely appreciate the reviewer's insightful comment regarding mechanistic explanations. In response to this valuable suggestion, we have significantly expanded our discussion to provide deeper insights into the underlying mechanisms driving our key observations. **For the Borderline stage's transitional nature**, we now discuss how its molecular plasticity may reflect competing adaptive (e.g., metabolic reprogramming) and maladaptive (e.g., early fibrotic) processes that create instability before NASH establishment (Line 461-467). **Regarding NASH interaction loss**, we elaborate on how chronic inflammation (via chemokine accumulation and MHC II upregulation) creates a hostile microenvironment that disrupts symbiotic host-microbe cross-talk (Line 482-487). **For mitophagy dysregulation**, we emphasize the sphingolipid-mediated mechanism through which *seq* influences mitochondrial quality control (Line 487-499).

The control was taken less (n= 72) where NAFL (n= 124), Borderline (n=143) and NASH (n= 231).

Answer: We acknowledge that equal sample sizes across all groups would be ideal for statistical comparisons. However, the considerable practical and ethical challenges in obtaining healthy liver tissue samples, necessarily limits our control group size. Despite the relatively smaller control group, it has met established requirements for transcriptomic studies, where groups of 40-50 typically provide sufficient power to detect biologically

meaningful differences²². In addition, our analytical approach incorporates stringent robust multiple testing corrections, and our results have maintained statistical significance. For example, after FDR correction, 1639, 1802, and 2407 genes (Figure 2a), as well as 18, 12, and 17 microbial species (Supplementary Table 8), remained significantly different in NAFL, Borderline, and NASH, compared to controls, respectively. Together, these provide strong evidence that the fewer sample size in control has not compromised our findings.

Fig 2. Host Gene Characteristics Associated with NAFLD Progression

a) Scatter plots illustrating the log₂ Fold Change (FC) values of genes (y-axis) between three disease stages vs. CTRL (control) (x-axis), red points represent differentially expressed genes (DEGs) whose FDR < 0.05 and absolute log₂FC > 1.2.

Supplementary Table 8: Differentially abundant hepatic microbes

NAFL vs Control			Borderline vs Control			NASH vs Control		
Name	Log2FC	FDR	Name	Log2FC	FDR	Name	Log2FC	FDR
Cloacibacterium normanense	-1.303	1.75E-11	Cloacibacterium normanense	-1.013	9.75E-07	Dolosigranulum pigrum	1.316	7.38E-12
Fulvia fulva	2.730	7.27E-10	Streptomyces sp. T12	2.142	3.98E-05	Streptomyces sp. T12	2.588	7.03E-09
Streptomyces sp. T12	2.971	7.27E-10	Methyloversatilis sp. RAC08	-1.012	7.16E-05	Cloacibacterium normanense	-1.016	7.03E-09
Cloacibacterium caeni	-1.345	9.04E-08	Stenotrophomonas maltophilia	2.028	3.08E-04	Methyloversatilis sp. RAC08	-1.256	7.14E-09
Plantactinospora sp. BCl	-2.694	1.20E-07	Plantactinospora sp. BBI	-1.726	3.08E-04	Cloacibacterium caeni	-1.076	2.80E-06
Plantactinospora sp. BBI	-2.296	2.43E-07	Plantactinospora sp. BCl	-1.920	4.33E-04	Malassezia vespertilionis	-1.719	1.12E-05
Methyloversatilis sp. RAC08	-1.217	2.43E-07	Malassezia vespertilionis	-1.568	5.54E-04	Plantactinospora sp. BBI	-1.767	1.67E-05
Stenotrophomonas maltophilia	2.632	3.78E-07	Ralstonia insidiosa	-1.573	1.70E-03	Cutibacterium acnes	-1.363	1.30E-04
Corynebacterium kefirresidentii	1.084	2.47E-05	Fulvia fulva	1.490	1.70E-03	Stenotrophomonas maltophilia	1.809	1.61E-04
Pseudomonas sp. CIP-10	1.742	4.96E-05	Staphylococcus epidermidis	-1.096	1.57E-02	Plantactinospora sp. BCl	-1.737	2.01E-04
Cutibacterium acnes	-1.514	1.11E-04	Acinetobacter baumannii	1.025	3.34E-02	Staphylococcus epidermidis	-1.413	2.12E-04
Acinetobacter baumannii	1.548	5.43E-04	Bacillus velezensis	1.416	4.09E-02	Acinetobacter johnsonii	1.089	3.33E-04
Moraxella osloensis	1.826	6.08E-04			Acinetobacter baumannii	1.387	5.38E-04	
Corynebacterium tuberculostearicum	1.184	2.88E-03			Rothia mucilaginosa	1.068	5.38E-04	
Escherichia coli	-1.481	4.79E-03			Ralstonia insidiosa	-1.270	2.69E-03	
Ralstonia insidiosa	-1.279	7.85E-03			Bacillus velezensis	1.645	4.60E-03	
Colletotrichum lupini	-1.088	1.65E-02			Escherichia coli	-1.048	2.79E-02	
Puccinia triticina	1.162	3.53E-02						

The Borderline stage needed more focus as it came out to be the transition state.

Answer: Our comprehensive analysis indeed reveals this phase as a critical transition point in NAFLD progression, characterized by distinct molecular and ecological features. We have substantially expanded our discussion of these Borderline characteristics in the revised manuscript (Line 461-470). We are grateful for this suggestion that has helped better highlight the transitional nature of this crucial disease phase.

The mechanistic explanation for interaction loss in NASH (vs. persistence in earlier stages) needs further expansion.

Answer: We sincerely appreciate the reviewer's insightful suggestion. The progression to NASH involves profound microenvironmental disruption characterized by sustained activation of pro-inflammatory pathways, including upregulated chemokine production and enhanced antigen presentation (highlighted with red frames in Figure 1d). We now expanded our discussion that this inflammatory milieu creates selective pressures that fundamentally reshape microbial communities, favoring the expansion of pathogenic species while depleting beneficial commensals like Sphingomonadaceae. In response, we have significantly expanded our discussion of this critical transition in the revised manuscript. (Line 482-487)

Fig 2. Host Gene Characteristics Associated with NAFLD Progression

d) Enrichment pathway plot showing the up-regulated pathways in the disease stage in each comparison with control, enriched by gene set enrichment analysis (GSEA). Each dot represents a pathway; the dot size correlates with the number of genes in the pathway. Similar pathways were grouped into clusters, each shown in the same background color. The most significant pathway in a cluster was chosen as the representative of that cluster.

There were lacking evidence in the study, confirming that the microbial changes can cause mitophagy impairment.

Answer: We sincerely appreciate the reviewer's insightful comment regarding the causal relationship between microbial changes and mitophagy impairment. Notably, our analysis has identified Sphingomonadaceae as a key microbial taxon that shows significant depletion in NASH compared to healthy controls (Figure S15). Interestingly, this species is known to produce sphingolipids, bioactive molecules that have been experimentally shown to regulate mitophagy. The observed reduction of this bacterial in NASH suggests a plausible mechanism through which microbial dysbiosis could contribute to mitophagy impairment via diminished sphingolipid signaling. We have supplemented the results and expanded the discussion section to more explicitly highlight both the potential significance of these findings. We also acknowledge the need for targeted experimental studies to establish causality, including fecal microbiota transplantation experiments in animal models and in vitro mechanistic studies (Line 402-410, 510-517).

Figure S15. Abundance of Sphingomonadaceae in different stages. Mean abundance of samples in each group was annotated in orange, p value was calculated by U test. *: p < 0.05

Future studies should explore whether specific microbial interventions can enhance mitochondrial quality control in liver disease.

Answer: We sincerely appreciate the reviewer's insightful suggestion regarding the potential of microbial interventions in liver disease. In response, we have expanded our discussion to highlight the therapeutic implications of targeting specific microbial taxa—particularly Sphingomonadaceae—given its crucial role in maintaining mitophagy through sphingolipid-mediated signaling (Line 490-499). Our revised manuscript now emphasizes that future studies should explore whether restoring Sphingomonadaceae abundance—via probiotics, prebiotics, or microbiota transplantation—could enhance mitochondrial quality control in NAFLD/NASH (Line 503-509). Given its dual role in mitophagy induction and immune modulation, this bacterium represents a promising candidate for interventions aimed at reducing ROS accumulation, improving mitochondrial

function, and mitigating disease progression. Additional functional experiments are warranted to fully characterize its therapeutic potential and underlying mechanisms in NAFLD.

Reviewer #2 (Comments for the Author):

The title of the manuscript reflects the content of the manuscript.

Answer: Thanks.

Bacteria and some viruses lack a polyA tail in their mRNA. The polyA tail-based mRNA enrich method and pure RNA extraction (without DNA contamination)-based RNA-seq data should avoid bacterial reads. The RNA extraction methods: The TRIzol method gives pure RNA with less DNA contamination, but the column-based RNA extraction has more DNA contamination. Therefore, the RNA extraction method and the library preparation method are also considered in the cohort analysis.

Answer: We sincerely appreciate the reviewer's thoughtful comments regarding potential technical considerations in sample processing. After carefully summarizing the experimental protocols across all five public datasets (Supplementary Table 1), we would like to provide the following clarifications regarding microbial detection and host transcriptomic profile in our study.

1. RNA Extraction Methods and Microbial Signal Preservation

Regarding RNA extraction methods, while four of the five cohorts used column-based approaches (RNeasy/AllPrep kits) and one employed magnetic bead-based isolation, it is important to note that **all methods, including magnetic bead-based isolation method, were designed to preserve total RNA, including microbial transcripts.** Furthermore, our bioinformatic processing were designed to **rigorously map these reads to transcriptome references.** Specifically, we first map all the sequencing reads to human transcriptome to remove host RNA signals. The unaligned reads were then subsequently mapped to customized reference database containing microbes, human and common contamination reads (downloaded from Refseq). These stringent approaches ensure that **the microbial reads we report represent authentic microbial-derived RNA rather than residual DNA.** However, we agree that further studies using RNA extraction methods optimized for purity could provide more unbiased observations and we have expanded our discussions (Line 527-530).

2. Library Preparation Strategies and Microbial Detection Consistency

Concerning library preparation strategies, our analysis incorporated both poly(A)+ enriched (four cohorts) and rRNA-depleted (US2 cohort) datasets. Although bacterial mRNAs lack stable poly(A) tails, microbial reads remain detectable in poly(A)+ libraries, likely because other type of bacterial RNAs, which constitutes 80–90% of prokaryotic RNA²³, is only partially removed during poly(A) enrichment²⁴. The validity of this approach is further supported by its successful application in multiple tissue microbiome studies²⁵⁻²⁹, demonstrating that residual microbial signals in poly(A)+ data yield biologically meaningful results. In addition, our direct comparison shows strong concordance ($r=0.99$, $p=3.49e-66$) between microbial profiles from different library types when analyzed using standardized pipelines (Figure S9). Another research has also observed a strong concordance between microbial profiles derived from poly(A)+ and rRNA-depleted data²⁴. In summary, **poly(A)+ selection does not completely eliminate microbial signals and library preparation method introduces minimal bias in our specific analytical context.**

We fully acknowledge the reviewer's broader point that tissue microbiome research would benefit from more rigorous experimental designs, particularly for capturing **non-polyadenylated microbial mRNA via random-primed libraries**. These limitations are now explicitly discussed in our revised manuscript, where we highlight future directions such as microbial RNA enrichment protocols, single-cell approaches, and combinatorial sequencing strategies to overcome current technical constraints (Line 517-522). While our study provides meaningful insights within the limitations of existing data, we agree that advancing this field will require methodological innovations to achieve more comprehensive and quantitative microbial profiling.

Supplementary Table 1. RNA extraction methods and library preparation strategies of cohorts.

Cohort	RNA Extraction Method	Extraction Type	Library Prep Kit	Library Type	Instrument
ER	AllPrep DNA/RNA Micro Kit	Column-based	TruSeq RNA Library Prep Kit v2	poly(A)+	NextSeq 550
US1	RNeasy Mini Kit	Column-based	NEBNext Poly(A) mRNA Magnetic Isolation Modul and NEBNext Ultra II Directional RNA Library Prep Kit	poly(A)+	Nova-Seq 6000
US2	MagMax AM1830 kit	Magnetic bead	TruSeq Stranded Total RNA LT Kit	rRNA-	HiSeq 3000
JP	RNeasy Plus Mini Kit	Column-based	TruSeq Stranded mRNA Kit	poly(A)+	HiSeq 3000
DM	RNeasy Mini Kit	Column-based	Ovation RNA-Seq System V2	poly(A)+	HiSeq 2000

Figure S9. High consistency of hepatic microbial abundance across different RNA library preparation methods.

Scatter plots showed that hepatic microbial abundance showed high agreement between polyA enrichment and rRNA depletion library preparation.

HiSeq, MiSeq, and NovaSeq are all sequencing instruments made by Illumina, and they also have differences in gene enrichment. Therefore, this platform of sequencers also needs to be considered in the cohort analysis.

Answer: We fully concur with the reviewer's critical assessment regarding the importance of addressing potential platform-specific biases in our multi-cohort analysis. To rigorously evaluate this concern, we performed systematic cross-platform validation across all Illumina sequencing systems (HiSeq, MiSeq, and NovaSeq) represented in our discovery cohorts. The results demonstrate remarkable technical reproducibility, with mean Spearman correlation coefficients exceeding 0.99 for host gene expression (Figure S6) and 0.97 for microbial abundance profiles (Figure S10). These findings provide strong evidence that platform differences did not significantly impact our results. We have incorporated these validation results in the revised manuscript (Line 290-292, 319-321).

Figure S6. High consistency of gene expression across different sequencing platforms.

Scatter plots showed that gene expression showed high agreement between NextSeq and HiSeq (a), NovaSeq and HiSeq (b), NextSeq and NovaSeq (c).

Figure S10. High consistency of hepatic microbial abundance across different sequencing platforms.

Scatter plots showed that hepatic microbial abundance showed high agreement between NextSeq and HiSeq (a), NovaSeq and HiSeq (b), NextSeq and NovaSeq (c).

In the case of differential gene expression, proper normalization needs to be used and justified for RNA extraction method, the library preparation method and the sequencing platform. The total amount of data (reads in GB) generated, the number of gene coverage in each sample, and the number of total reads after trimming in each sample also affect the normalized gene count from

different Biosample IDs.

Answer: We sincerely appreciate the reviewer's comments. In response, we have rigorously re-analyzed our data by incorporating these technical factors as covariates in our limma-based differential expression analysis: (1) gene coverage per sample, (2) number of clean reads after trimming, (3) sequencing platform, (4) RNA extraction method, and (5) library preparation method. We selected post-trimming read counts rather than total raw data volume because our gene was annotated based on those quality-filtered reads, and we believe low-quality trimmed reads should not influence the normalization process.

The results of this comprehensive re-analysis demonstrate remarkable consistency with our original findings. When comparing the adjusted results (with all technical covariates) to our initial analysis, we observed an extremely high correlation in log₂ fold changes (log₂FC) of gene expression across all disease stages (NAFL: $R^2 = 0.90$, $p < 0.05$; Borderline: $R^2 = 0.93$, $p < 0.05$, NASH: $R^2 > 0.99$, $p < 0.05$) relative to controls (Figure S7). This robust consistency strongly supports the reliability of our findings and suggests that while these technical factors are important to account for, they did not substantially alter our core biological conclusions.

We have incorporated these validation results into the revised manuscript (Lines 293-296). This additional rigor further strengthens the validity of our study's transcriptional profiling results. We thank the reviewer for this constructive suggestion that has enhanced the methodological robustness of our work.

Figure S7. The influence of adjusting covariates to gene expression fold change of diseased stages compared to healthy controls.

Scatter plots showed that, compared to the healthy control, the fold change of gene expression in NAFL (a), Borderline (b) and NASH (c) remained stable after adjusting for covariates including gene coverage per sample, number of clean reads after trimming, sequencing platform, RNA extraction method, and library preparation method.

The biopsy procedure and part of the liver samples also affect gene expression. How the control samples were collected, if autopsy samples, how many hours after death the samples were collected, also affects the gene expression.

Answer: Through thorough examination of the original study protocols and direct

communication with corresponding authors, we confirmed that all biopsy samples in our cohorts were obtained **through standardized percutaneous needle biopsy procedures, with the majority collected from the right lobe of the liver.** The control samples were obtained either **during bariatric surgery or other non-hepatic surgical procedures,** providing metabolically relevant comparator tissues. We have modified our manuscript to clarify about these procedure details (Line 144-145).

How many bacteria are identified, how many genes of each bacterium are identified, what is the death coverage of each bacterial gene, how many reads per bacterial gene before and after normalization, what is the genome coverage of each bacterial gene, and the ORF nucleotide sequences details are not provided in this manuscript. Any antibiotic-resistant ORF identified. After these analyses, one can conclude the findings of this manuscript.

Answer: We sincerely appreciate the reviewer's insightful suggestions regarding bacterial gene-level analysis, which indeed represents an important direction for future research. While our primary focus remains on understanding how microbial communities may influence host transcriptomic activity, we fully recognize that obtaining detailed information on bacterial gene coverage, ORF sequences, and normalized read counts would provide valuable insights into direct bacterial function and have framed this as a key future direction.

Currently, the characterization of tissue-associated microbiota presents unique technical challenges due to their exceptionally low biomass, with bacterial transcripts typically constituting less than 1% of total RNA content²⁹. This extremely low abundance makes traditional metagenomic assembly and gene-level quantification particularly challenging, as it often results in insufficient coverage for reliable genome assembly or ORF-level analysis. While our k-mer based approach using KrakenUniq³⁰, provides robust taxonomic profiling in such low-biomass scenario through unique k-mer signatures and stringent false-positive filters, this alignment-free method cannot resolve gene-specific information due to the absence of complete genomic contigs. However, we completely agree with the reviewer that future studies employing microbial RNA enrichment or ultra-deep sequencing could bridge this gap by enabling direct functional characterization of tissue-associated microbiota. Relevant discussion was modified in our manuscript (Line 517-524, 527-529).

Reference:

1. Liu, B.M., and Hayes, A.W. (2024). Mechanisms and Assessment of Genotoxicity of Metallic Engineered Nanomaterials in the Human Environment. *Biomedicines* 12, 2401. 10.3390/biomedicines12102401.
2. García-Ruiz, C., and Fernández-Checa, J.C. (2018). Mitochondrial Oxidative Stress and Antioxidants Balance in Fatty Liver Disease. *Hepatology* 67, 1425-1439. 10.1002/hep4.1271.
3. Nguyen, G.T., Green, E.R., and Meccas, J. (2017). Neutrophils to the ROScue: Mechanisms of NADPH Oxidase Activation and Bacterial Resistance. *Frontiers in Cellular and Infection Microbiology* 7. 10.3389/fcimb.2017.00373.
4. Bonekamp, N.A., Völkl, A., Fahimi, H.D., and Schrader, M. (2009). Reactive oxygen species and peroxisomes: Struggling for balance. *BioFactors* 35, 346-355. 10.1002/biof.48.

5. Fleming, I., Michaelis, U.R., Bredenkötter, D., Fisslthaler, B., Dehghani, F., Brandes, R.P., and Busse, R. (2001). Endothelium-Derived Hyperpolarizing Factor Synthase (Cytochrome P450 2C9) Is a Functionally Significant Source of Reactive Oxygen Species in Coronary Arteries. *Circulation Research* 88, 44-51. 10.1161/01.RES.88.1.44.
6. Yang, W., Tao, Y., Wu, Y., Zhao, X., Ye, W., Zhao, D., Fu, L., Tian, C., Yang, J., He, F., and Tang, L. (2019). Neutrophils promote the development of reparative macrophages mediated by ROS to orchestrate liver repair. *Nature Communications* 10, 1076. 10.1038/s41467-019-09046-8.
7. Balaban, R.S., Nemoto, S., and Finkel, T. (2005). Mitochondria, oxidants, and aging. *Cell* 120, 483-495. 10.1016/j.cell.2005.02.001.
8. Zorov, D.B., Juhaszova, M., and Sollott, S.J. (2014). Mitochondrial Reactive Oxygen Species (ROS) and ROS-Induced ROS Release. *Physiological Reviews* 94, 909-950. 10.1152/physrev.00026.2013.
9. Ma, X., McKeen, T., Zhang, J., and Ding, W.-X. (2020). Role and Mechanisms of Mitophagy in Liver Diseases. *Cells* 9, 837. 10.3390/cells9040837.
10. Che, Z., Zhou, Z., Li, S.-Q., Gao, L., Xiao, J., and Wong, N.-K. (2023). ROS/RNS as molecular signatures of chronic liver diseases. *Trends in Molecular Medicine* 29, 951-967. 10.1016/j.molmed.2023.08.001.
11. Mitochondrial oxidative function in NAFLD: Friend or foe? (2021). *Molecular Metabolism* 50, 101134. 10.1016/j.molmet.2020.101134.
12. Schwenger, K.J.P., Copeland, J.K., Ghorbani, Y., Chen, L., Comelli, E.M., Guttman, D.S., Fischer, S.E., Jackson, T.D., Okrainec, A., and Allard, J.P. (2025). Characterization of liver, adipose, and fecal microbiome in obese patients with MASLD: links with disease severity and metabolic dysfunction parameters. *Microbiome* 13, 9. 10.1186/s40168-024-02004-7.
13. Liu, B.M., Beck, E.M., and Fisher, M.A. (2021). The Brief Case: Ventilator-Associated *Corynebacterium accolens* Pneumonia in a Patient with Respiratory Failure Due to COVID-19. *Journal of Clinical Microbiology* 59, e0013721. 10.1128/JCM.00137-21.
14. Ridaura, V.K., Bouladoux, N., Claesen, J., Chen, Y.E., Byrd, A.L., Constantinides, M.G., Merrill, E.D., Tamoutounour, S., Fischbach, M.A., and Belkaid, Y. (2018). Contextual control of skin immunity and inflammation by *Corynebacterium*. *Journal of Experimental Medicine* 215, 785-799. 10.1084/jem.20171079.
15. Li, M., Kopylova, E., Mao, J., Namkoong, J., Sanders, J., and Wu, J. (2024). Microbiome and lipidomic analysis reveal the interplay between skin bacteria and lipids in a cohort study. *Frontiers in Microbiology* 15. 10.3389/fmicb.2024.1383656.
16. Sheng, Q., Vickers, K., Zhao, S., Wang, J., Samuels, D.C., Koues, O., Shyr, Y., and Guo, Y. (2017). Multi-perspective quality control of Illumina RNA sequencing data analysis. *Briefings in Functional Genomics* 16, 194-204. 10.1093/bfpg/ew035.
17. Loomba, R., Friedman, S.L., and Shulman, G.I. (2021). Mechanisms and disease consequences of nonalcoholic fatty liver disease. *Cell* 184, 2537-2564. 10.1016/j.cell.2021.04.015.
18. Tilg, H., Adolph, T.E., and Trauner, M. (2022). Gut-liver axis: Pathophysiological concepts and clinical implications. *Cell Metabolism* 34, 1700-1718. 10.1016/j.cmet.2022.09.017.
19. Bauer, K.C., Littlejohn, P.T., Ayala, V., Creus-Cuadros, A., and Finlay, B.B. (2022). Nonalcoholic Fatty Liver Disease and the Gut-Liver Axis: Exploring an Undernutrition Perspective. *Gastroenterology* 162, 1858-1875.
20. Champion, C. (2023). Human liver microbiota modeling strategy at the early onset of fibrosis. *BMC Microbiology* 23, 34.

21. Sookoian, S., Salatino, A., Castaño, G.O., Landa, M.S., Fijalkowky, C., Garaycochea, M., and Pirola, C.J. (2020). Intrahepatic bacterial metataxonomic signature in non-alcoholic fatty liver disease. *Gut* *69*, 1483-1491. 10.1136/gutjnl-2019-318811.
22. Zhao, S., Li, C.-I., Guo, Y., Sheng, Q., and Shyr, Y. (2018). RnaSeqSampleSize: real data based sample size estimation for RNA sequencing. *BMC Bioinformatics* *19*, 191. 10.1186/s12859-018-2191-5.
23. Westermann, A.J., and Vogel, J. (2021). Cross-species RNA-seq for deciphering host-microbe interactions. *Nature Reviews Genetics* *22*, 361-378. 10.1038/s41576-021-00326-y.
24. Hu, X., Haas, J.G., and Lathe, R. (2022). The electronic tree of life (eToL): a net of long probes to characterize the microbiome from RNA-seq data. *BMC Microbiology* *22*, 317. 10.1186/s12866-022-02671-2.
25. Sepich-Poore, G.D., McDonald, D., Kopylova, E., Guccione, C., Zhu, Q., Austin, G., Carpenter, C., Fraraccio, S., Wandro, S., Kosciolk, T., et al. (2024). Robustness of cancer microbiome signals over a broad range of methodological variation. *Oncogene* *43*, 1127-1148. 10.1038/s41388-024-02974-w.
26. Luo, M., Liu, Y., Hermida, L.C., Gertz, E.M., Zhang, Z., Li, Q., Diao, L., Rupp, E., and Han, L. (2022). Race is a key determinant of the human intratumor microbiome. *Cancer Cell* *40*, 901-902. 10.1016/j.ccell.2022.08.007.
27. Lu, L., Johnson, C., Khan, S., and Kluger, H. (2024). 16S rRNA target sequencing of human tumors validates findings of Lachnospirillum abundance in human melanomas that are heavily CD8+ T-cell infiltrated. *European Journal of Cancer* *213*. 10.1016/j.ejca.2024.115084.
28. Hermida, L.C., Gertz, E.M., and Rupp, E. (2022). Predicting cancer prognosis and drug response from the tumor microbiome. *Nature Communications* *13*, 2896. 10.1038/s41467-022-30512-3.
29. Hadzega, D., Minarik, G., Karaba, M., Kalavska, K., Benca, J., Ciernikova, S., Sedlackova, T., Nemcova, P., Bohac, M., Pindak, D., et al. (2021). Uncovering Microbial Composition in Human Breast Cancer Primary Tumour Tissue Using Transcriptomic RNA-seq. *International Journal of Molecular Sciences* *22*, 9058. 10.3390/ijms22169058.
30. Breitwieser, F.P., Baker, D.N., and Salzberg, S.L. (2018). KrakenUniq: confident and fast metagenomics classification using unique k-mer counts. *Genome Biology* *19*, 1-10. 10.1186/s13059-018-1568-0.

Re: Spectrum00100-25R1 (**Disrupted Host-Microbiota Crosstalk Promotes Nonalcoholic Fatty Liver Disease Progression by Impaired Mitophagy**)

Dear Prof. Na Jiao:

Your manuscript has been accepted, and I am forwarding it to the ASM production staff for publication. Your paper will first be checked to make sure all elements meet the technical requirements. ASM staff will contact you if anything needs to be revised before copyediting and production can begin. Otherwise, you will be notified when your proofs are ready to be viewed.

Sincerely,
Benjamin Liu
Editor
Microbiology Spectrum